# GANs Class-Conditioning Methods: A Survey

## Abstract

In recent years, Generative Adversarial Networks (GANs) have seen significant advancements, leading to their widespread adoption across various fields. The original GAN architecture enables the generation of images without any specific control over the content, making it an unconditional generation process. However, many practical applications require precise control over the generated output, which has led to the development of conditional GANs (cGANs) that incorporate explicit conditioning to guide the generation process. cGANs extend the original framework by incorporating additional information (conditions), enabling the generation of samples that adhere to that specific criteria. Various conditioning methods have been proposed, each differing in how they integrate the conditioning information into both the generator and the discriminator networks. In this work, we review the conditioning methods proposed for GANs, exploring the characteristics of each method and highlighting their unique mechanisms and theoretical foundations. Furthermore, we conduct a comparative analysis of these methods, evaluating their performance on various image datasets. Through these analyses, we aim to provide insights into the strengths and limitations of various conditioning techniques, guiding future research and application in generative modeling.

## 1 Introduction

### 1.1 Generative Adversarial Networks: An overview

Generative Adversarial Networks (GANs) (Goodfellow et al., 2014) are implicit generative model in which the data distribution is learned by comparing real samples with generated ones. This approach leverages an adversarial process between two neural networks: a generator, which produces synthetic data, and a discriminator, which evaluates the data to distinguish between real and generated (fake) samples. The competition between these networks drives both to improve, with the generator aiming to create increasingly realistic data while the discriminator becomes better at identifying fakes. Since their introduction, GANs have inspired numerous extensions and enhancements. Notably, the Deep Convolutional GAN (DCGAN) (Radford et al., 2016), marked a significant advancement by employing convolutional layers. DCGAN demonstrated the ability to generate high-quality images and contributed to the robustness of GAN training. To address the inherent training difficulties and instability of GANs, different objectives were proposed (Arjovsky et al., 2017; Gulrajani et al., 2017; Mroueh et al., 2017a;b; Li et al., 2017), leading to more stable training and ultimately producing higher quality outputs. Self-Attention GANs (SAGAN) (Zhang et al., 2019) enhanced GANs' ability to capture global dependencies within images by integrating self-attention mechanisms (Vaswani et al., 2023). BigGAN (Brock et al., 2019), scaled up the GAN architecture, achieving spectacular results on the ImageNet dataset through large batch sizes and careful architectural choices, pushing the boundaries of what GANs can achieve in terms of image quality and diversity. Furthermore, (Karras et al., 2018) proposed the Progressive Growing GAN (ProGAN), a method to train GANs starting with low-resolution images and incrementally increasing the resolution as training progresses. This approach mitigated training instability and produced unprecedentedly high-resolution images. Additionally, StyleGAN (Karras et al., 2019), introduced a style-based generator that allows for fine-grained control over the generated image features, setting new benchmarks in image synthesis. These foundational works have collectively expanded the application of GANs, including but not limited to image synthesis, data augmentation (Motamed et al.,

2021), super-resolution (Tian et al., 2022), and even creative domains such as art generation (Shahriar, 2022).

## 2 Conditional GANs

Controlling the generative process of images is crucial for many applications such as image editing (Gauthier, 2015; Antipov et al., 2017; Perarnau et al., 2016; Gu et al., 2019), text-based image generation (Reed et al., 2016b; Hong et al., 2018; Zguimg et al., 2017; Shi et al., 2024), 3D scene manipulation (Yao et al., 2018), time series analysis (Smith & Smith, 2020), medical imaging (Havaei et al., 2021; Dar et al., 2019; Bourou et al., 2023) and audio generation (Kong et al., 2020; Zeghidour et al., 2021; Défossez et al., 2022). Although cGANs were initially mentioned as a straightforward extension of GANs (Goodfellow et al., 2014), they were first formally introduced in (Mirza & Osindero, 2014). Their approach introduced conditioning by incorporating a class embedding variable, which was concatenated with the input data fed into both the generator and the discriminator. The Auxiliary Classifier GAN (AC-GAN) (Shu, 2017) advanced this concept by adding an auxiliary classifier to the discriminator, enabling it to distinguish between different classes of images. While AC-GAN improved the quality of conditional generation, it tended to produce images that were easy to classify (Shu, 2017). Moreover, as the number of classes increased, AC-GAN was prone to early collapse during the training process (Shu, 2017; Han et al., 2020).

To address these issues, several subsequent works introduced significant improvements. In (Miyato & Koyama, 2018) proposed to perform the conditioning using a projection-based discriminator, achieving remarkable synthesis results, a principle later adopted by other cGAN architectures such as StyleGANs (Karras et al., 2019; 2020b; 2021) and BigGAN (Brock et al., 2019). ContraGAN (Kang & Park, 2021) used contrastive loss to better capture data-to-data relations, improving the conditioning process. Although recent advancements in other generative models, such as diffusion models (Ho et al., 2020), have gained significant attention, GANs continue to undergo major developments (Huang et al., 2024), making them competitive models with valuable properties (see Appendix A).

Despite the numerous advancements and variations in conditional GANs, no prior surveys have comprehensively discussed the different architectures used in class-conditional GANs. This work aims to fill this gap by providing an extensive overview of the methods used to condition GANs, comparing their effectiveness, and evaluating their robustness and performance.

The adversarial training scheme for GANs (Goodfellow et al., 2014) is described by the following loss function:

$$\min_G \max_D \mathbb{E}_{x \sim p(x)}[\log D(x)] + \mathbb{E}_{z \sim p_z(z)}[\log (1 - D(G(z)))] \tag{1}$$

Where $G$ and $D$ are, respectively, a parameterized generator and discriminator, $p(x)$ is the real data distribution, and $p_z(z)$ is a multivariate random distribution.

GANs can be made conditional by considering additional information such as class labels, text, image, or other modalities. While conditioned, the GAN objective1 can be reformulated as:

$$\min_G \max_D \mathbb{E}_{x \sim p(x)}[\log D(x|y)] + \mathbb{E}_{z \sim p(z)}[\log (1 - D(G(z|y)))] \tag{2}$$

In this survey, we focus on works that use class labels for conditioning, although many of the techniques discussed can be extended to other modalities. Conditioning GANs involves applying conditioning to both the generator and the discriminator. To address this systematically, we have organized our work as follows: In Section 2, we outline the strategies proposed for conditioning the discriminator. Next, in Section 3, we describe the techniques suggested for conditioning the generator. Finally, we provide a comparison to evaluate and contrast the effectiveness of the proposed conditional GAN approaches.

## 3 Conditional GAN: conditioning the discriminator

The discriminator in GANs plays a crucial role, by providing feedback on the quality of the generated data samples to the generator. In the conditional setting, the discriminator should be provided with the class label, the earliest cGANs frameworks fed the class label $y$ to the discriminator by simply concatenating it with the feature vector (Mirza & Osindero, 2014). Variants of this method have proposed concatenating the class label embedding with the feature vector at different depths in the network (Kwak & Zhang, 2016; Zguimg et al., 2017; Saito et al., 2017; Dumoulin et al., 2017a; Perarnau et al., 2016; Denton et al., 2015; Reed et al., 2016a).

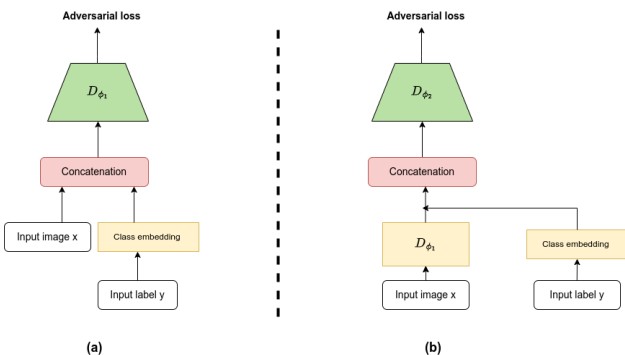

Figure 1: Conditioning by label embedding concatenation at different levels of the discriminator, (a) was proposed in (Mirza & Osindero, 2014) and (b) in (Reed et al., 2016a)

Subsequent works introduced new techniques for injecting the class label information into the discriminator. Depending on the discriminator conditioning method, we can group the cGAN discriminators into: **auxiliary classifier-based discriminators**, **projection-based discriminators** and **constrastive learning-based discriminators**. In the subsequent sections, we will delve into each method, outlining their respective mechanisms, in Table 3, we summarize the different methods, their losses and their schematics.

### 3.1 Auxiliary-classifier based discriminators

Concatenating class label information with the input image features can achieve conditioning; however, this approach is relatively simplistic and arbitrary, which may hinder GANs from accurately approximating the true data distribution. In this section, we present the methods that condition the discriminator by incorporating an auxiliary classifier.

#### 3.1.1 Auxiliary classifier GAN (AC-GAN)

The development of conditional Generative Adversarial Networks (cGANs) advanced significantly with the introduction of **Auxiliary Classifier GANs (AC-GANs)** (Odena et al., 2017). This approach integrates an auxiliary classifier into the GAN discriminator to predict the class label of the generated image. This design shift, motivated by the potential for enhanced performance through multi-task learning, enables the AC-GAN to generate higher quality and class-specific images. Unlike prior cGANs, where conditioning information is directly fed to the discriminator via concatenation, AC-GAN employs a dual objective function. The first part, $L_S$ (Eq. 3), focuses on the log-likelihood of correctly identifying real versus fake images, while the second part, $L_C$ (Eq. 4), concentrates on accurately classifying these images into their respective classes:

$$L_S = \mathbb{E}_{x \sim p(x)}[\log D(x)] + \mathbb{E}_{z \sim p_z(z), y \sim p_y(y)}[\log (1 - D(G(z, y)))] \tag{3}$$

$$L_C = -\mathbb{E}_{x \sim p(x,y)}[\log C(x, y)] - \mathbb{E}_{z \sim p_z(z), y \sim p_y(y)}[\log(C(G(z, y)), y)] \tag{4}$$

Where $C$ is the introduced auxiliary classifier, By combining $L_S$ and $L_C$ we obtain the AC-GAN loss:

$$
\begin{aligned}
\min_{G,C} \max_D \mathcal{L}_{AC}(G, D, C) = & \mathbb{E}_{x \sim p_x(x)}[\log D(x)] + \mathbb{E}_{z \sim p_z(z), y \sim p_y(y)}[\log(1 - D(G(z, y)))] \\
& - \lambda_c \mathbb{E}_{x \sim p(x,y)}[\log C(x, y)] - \lambda_c \mathbb{E}_{z \sim p_z(z), y \sim p_y(y)}[\log(C(G(z, y), y))]
\end{aligned}
\tag{5}
$$

where $\lambda_C$ is a hyperparameter.

Despite its advancements, **AC-GAN** suffers from a lack of diversity in the generated images, particularly as the number of classes increases. This issue arises from the model's tendency to generate images that are easier for the auxiliary classifier to categorize. This bias was explored in (Shu, 2017), where **AC-GAN** was described as the Lagrangian of a constrained optimization problem that rejects the sampling of points near the classifier decision boundaries.

### 3.1.2 Twin Auxiliary Classifier GAN (TAC-GAN)

To address the limitations of **AC-GAN**, the **Twin Auxiliary Classifier GAN (TAC-GAN)** was proposed by (Gong et al., 2019), introducing an additional auxiliary classifier. In (Gong et al., 2019), it was shown that the absence of the negative conditional entropy term $-H_q(y|x)$ in the objective function of **AC-GAN** can lead to a degenerate solution that causes the generated images to be confined by the decision boundaries of the auxiliary classifier. This behavior explains the low intra-class diversity observed in the images synthesized by **AC-GAN**. To alleviate this issue, the authors proposed to add an additional classifier to the **AC-GAN** that predicts the class of the generated images. This additional auxiliary classifier $C^{mi}$ is trained to compete with the generator, optimizing the following objective function:

$$
\min_G \max_{C^{mi}} V(G, C^{mi}) = \mathbb{E}_{z \sim p_z(z), y \sim p_y(y)}[\log C^{mi}(G(z, y), y)]
\tag{6}
$$

combining Eq. 6 with the original AC-GAN objective leads to the total loss of TAC-GAN that reads:

$$
\min_{G,C} \max_{D,C^{mi}} \mathcal{L}_{TAC}(G, D, C, C^{mi}) = \mathcal{L}_{AC}(G, D, C) + \lambda_{ac} V(G, C^{mi})
\tag{7}
$$

The authors established a connection between the conditional entropy term and the Jensen-Shannon Divergence (JSD) among the conditional distributions $\{q_{x|y=1}, \ldots, q_{x|y=K}\}$. Furthermore, they demonstrated that **TAC-GAN**'s loss function minimizes the JSD between these conditional distributions. This indicates that **TAC-GAN** effectively addresses the issue of the missing conditional entropy term. Additionally, the proposed loss function proves to be beneficial in learning an unbiased distribution and generating more diverse images.

### 3.1.3 Unbiased Auxiliary Classifier GAN (UAC-GAN)

Similar to (Gong et al., 2019), (Han et al., 2020) demonstrated that the lack of diversity observed in **AC-GAN** is induced by the absence of $-H_q(y|x)$ in the AC-GAN objective function. Furthermore, it was shown that **TAC-GAN** can still converge to a degenerate solution. In addition to that, it was observed that using an additional classifier can lead to an unstable training (Kocaoglu et al., 2017; Han et al., 2020). Instead of using an additional classifier to minimize $-H_q(y|x)$, (Han et al., 2020) proposed to estimate the mutual information $I_q(x; y)$ since:

$$
I_q(x; y) = H_q(y) - H_q(y|x) = H_q(x) - H_q(x|y)
\tag{8}
$$

To estimate $I_q(x; y)$, they employed the *Mutual Information Neural Estimator(MINE)* (Belghazi et al., 2021). MINE is built on top of the Donsker and Varadhan bound (Donsker & Varadhan, 1975), $I_Q$ can be estimated using the following equation:

$$
I_q^{MINE}(x, y) = \max_T V_{MINE}(G, T)
\tag{9}
$$

Table 1: learning objective for the generator under the optimal discriminator and classifier.

| Method | Learning Objective for the Generator |
|--------|--------------------------------------|
| AC-GAN | $\min\limits_{G} JS(p_x\|q_x) + \lambda_1 KL(q_{x,y}\|p_{x,y})) - KL(q_x\|p_x)) + H_q(y\|x))$ |
| TAC-GAN | $\min\limits_{G} JS(p_x\|q_x)) + \lambda_1 KL(q_{x,y}\|p_{x,y})) - KL(q_x\|p_x))$ |
| ADC-GAN | $\min\limits_{G} JS(p_x\|q_x) + \lambda_1 KL(q_{x,y}\|p_{x,y})$ |

where:

$$V_{MINE}(G,T) = \mathbb{E}_{z\sim p(z),y\sim p(y)}[T(G(z,y),y)] - \log \mathbb{E}_{z\sim p(z),y\sim q(y)} \exp\left(T(G(z,y),y\right) \tag{10}$$

$T$ is a scalar-valued function that can be parameterized by a deep neural network. The final objective function is given by:

$$\min_{G,C} \max_{D,T}(G,D,C,T) = L_{AC}(G,D,C) + \lambda_m V_{MINE}(G,T) \tag{11}$$

where $\lambda_m$ is a hyperparameter.

By linking $-H_q(y|x)$ to mutual information, (Han et al., 2020) demonstrated an effective approach to addressing the lack of diversity in **AC-GAN** without requiring an additional classifier, which often causes unstable training.

### 3.1.4 Auxiliary Discriminative Classifier GAN (ADC-GAN)

In another work, (Hou et al., 2021) proposed **Auxiliary Discriminative Classifier GAN (ADC-GAN)** to overcome the limitations of **AC-GAN**. They demonstrated that, for a fixed generator, the optimal classifier of **AC-GAN** is agnostic to the density of the generated distribution $q(x)$. Furthermore, they highlighted that the generators in **TAC-GAN** and **AC-GAN** optimize contradictory learning objectives as shown in Table 1.

To alleviate these shortcomings, **ADC-GAN** uses a classifier that is able to classify the real data and the generated data separately. Using such a classifier $C_d : X \to Y^+ \cup Y^-$ ($Y^+$ for real data and $Y^-$ for generated data), the generator is encouraged to produce classifiable samples that look like the real ones. The objective functions for the discriminator, the discriminative classifier and the generator are:

$$\max_{D,C_d} V_{AC}(G,D) + \lambda(\mathbb{E}_{x,y\sim p_{x,y}}[\log C_d(y^+|x)] + \mathbb{E}_{x,y\sim q_{x,y}}[\log C_d(y^-|x)])$$
$$\min_{G} V_{AC}(G,D) - \lambda(\mathbb{E}_{x,y\sim p_{x,y}}[\log C_d(y^+|x)] - \mathbb{E}_{x,y\sim q_{x,y}}[\log C_d(y^-|x)]) \tag{12}$$

where:

$C_d(y^+|x) = \frac{\exp(\varphi^+(y)\cdot\phi(x))}{\sum_{\bar{y}}\exp(\varphi^+(\bar{y})\cdot\phi(x))+\sum_{\bar{y}}\exp(\varphi^-(\bar{y})\cdot\phi(x))}$ and $C_d(y^-|x) = \frac{\exp(\varphi^-(y)\cdot\phi(x))}{\sum_{\bar{y}}\exp(\varphi^+(\bar{y})\cdot\phi(x))+\sum_{\bar{y}}\exp(\varphi^-(\bar{y})\cdot\phi(x))}$

The function $\phi : \mathcal{X} \to \mathbb{R}^d$ serves as a feature extractor, transforming input data $X$ into a $d$-dimensional feature space. This feature extractor is shared with the original discriminator, which is represented as $D = \sigma \circ \psi \circ \phi$. Here, $\psi : \mathbb{R}^d \to \mathbb{R}$ is a linear mapping, and $\sigma : \mathbb{R} \to [0,1]$ is a sigmoid function. Additionally, $\varphi^+ : \mathcal{Y} \to \mathbb{R}^d$ and $\varphi^- : \mathcal{Y} \to \mathbb{R}^d$ are learnable embeddings capturing the label representations for real and generated data, respectively and $V_{AC}$ is the the original loss for **AC-GAN**.

The authors of ADC-GAN proved that for a fixed generator, the optimal discriminative classifier is given as: $C_d^*(y^+|x) = \frac{p(x,y)}{p(x)+q(x)}$, $C_d^*(y^-|x) = \frac{q(x,y)}{p(x)+q(x)}$ which shows that the optimal discriminative classifier is aware of the real and generated densities. Furthermore, they were able to discard the $-KL(p_x|q_x)$ term from the objective of the generator as shown in Table 1 In addition, they demonstrated that **ADC-GAN** leads to a

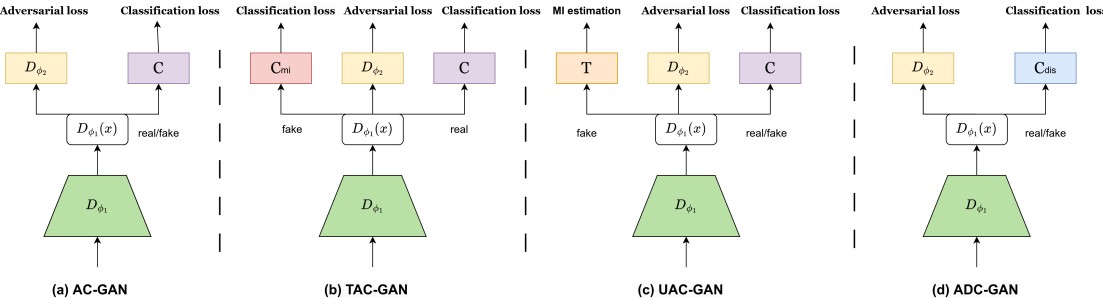

Figure 2: Auxiliary classifier based architectures: (a) AC-GAN, (b) TAC-GAN, (c) UAC-GAN, (d) ADC-GAN

more stable training and outperforms **AC-GAN** and **TAC-GAN** in terms of the quality of the generated images.

## 3.2 Projection-based discriminators

In the previous sections, conditioning the discriminator was achieved either by concatenating the class label or by adding an auxiliary classifier. Where the former can method can be very naive and sub-optimal in capturing the additional class label information, the latter can make the training more difficult and unstable. In (Miyato & Koyama, 2018), a new method for conditioning the discriminator was introduced by computing the inner product between the embedded conditional vector $y$ and the feature vector.

The design introduced in (Miyato & Koyama, 2018) presents a novel method for cGANs by employing a projection discriminator, it was proposed by considering the optimal solution for the discriminator's loss function, (Miyato & Koyama, 2018) demonstrated that under certain regularity assumptions, the discriminator's function can be reparameterized as follows:

$$f(x,y;\theta) = f_1(x,y;\theta) + f_2(x;\theta) = y^T V \phi(x;\theta_\Phi) + \psi(\phi(x;\theta_\Phi);\theta_\Psi) \tag{13}$$

where $V$ is the embedding matrix of $y$, $\phi(.,\theta_\Phi)$ is a vector output function of $x$, and $\psi(.,\theta_\Psi)$ is a scalar function. The learned parameters $\theta = \{V, \theta_\phi, \theta_\psi\}$ are trained to optimize the adversarial loss.

The projection discriminator approach for conditional Generative Adversarial Networks (cGANs), as proposed in (Miyato & Koyama, 2018) offers notable improvements over traditional methods like concatenation. This technique enhances inter-class diversity, producing more varied and realistic samples across different classes, which is crucial in many applications. A significant advantage of this method is its avoidance of additional classifiers, leading to a greater training stability. The effectiveness and versatility of this approach are further evidenced by its adoption in various advanced GAN architectures, as seen in (Brock et al., 2019; Zhang et al., 2020; Zhao et al., 2020b; Wu et al., 2020).

BigGAN (Brock et al., 2019), was among the pioneering GANs to employ discriminator projection techniques for conditional generation. It brought significant enhancements to the scaling of GAN training, enabling the generation of images with higher resolutions. A pivotal enhancement in BigGAN's design is the integration of orthogonal regularization, which contributed markedly to its improved performance. Furthermore, BigGAN drew inspiration from the Self-Attention GAN (Zhang et al., 2019), particularly its utilization of self-attention blocks. These blocks aid both the discriminator and generator in more effectively capturing the global structure of images. Additionally, BigGAN's architecture facilitated the application of the truncation trick, which allows for nuanced balancing of the fidelity-diversity trade-off in generated images.

Another line of work that adopted the projection discriminator is the StyleGAN (Karras et al., 2019; 2020b; 2021). The StyleGAN family of models represents a significant advancement in the use of projection discriminators. These models have achieved new state of the art results by incorporating innovative components

like the Mapping Network and AdaIN normalization (Huang & Belongie, 2017). Moreover, several techniques were introduced to enhance the quality of image generation, even when dealing with limited size data sets (Huang & Belongie, 2017; Zhao et al., 2020a).

### 3.3 Contrastive learning based discriminators

Contrastive learning (Chen et al., 2020; Jaiswal et al., 2021; Le-Khac et al., 2020) mainly aims to develop deep, meaningful, and robust data representations. At its core, it involves training models to distinguish between pairs of examples that are either similar or dissimilar. During the training phase, the model is encouraged to draw closer the representations of similar items ('positive' pairs) while distancing those of dissimilar items ('negative' pairs). This approach not only strengthens the model's capacity to discern underlying data structures and patterns but also enhances generalization across various tasks. The effectiveness of contrastive learning is particularly evident in diverse domains such as computer vision (He et al., 2021; 2020; Addepalli et al., 2022) and text processing (Gunel et al., 2021; Chen et al., 2023; Aberdam et al., 2020). More recently, its application has been extended to generative models, as explored in (Kang & Park, 2021; Kang et al., 2021), demonstrating its versatility and growing importance in the field of machine learning.

#### 3.3.1 Contrastive learning GAN (ContraGAN)

ContraGAN (Kang & Park, 2021) is a cGAN that achieve conditioning using a contrastive learning strategy by capturing the data-to-data relations. Indeed, it was suggested in (Kang & Park, 2021) that the conditioning in **AC-GAN** and **ProjGAN** can only capture data-to-class relations of training examples while neglecting the data-to-data relations. To alleviate this, (Kang & Park, 2021) have proposed the **Conditional Contrastive (2C) loss**, a self-supervised learning objective that controls the distances between embedded images depending on their respective labels.

The 2C loss can be seen as an adaptation of the NT-Xent loss (Chen et al., 2020). Given a minibatch of training images $\mathbf{X} = \{\mathbf{x}_1, \ldots, \mathbf{x}_m\}$, where $x \in \mathbb{R}^{W \times H \times 3}$ and their corresponding labels $\mathbf{y} = \{y_1, \ldots, y_m\}$, an encoder $S(x) \in \mathbb{R}^k$, a projection layer $h : \mathbb{R}^k \to \mathbb{S}^d$ that embeds onto a unit hypersphere, the NT-Xent loss conducts random data augmentations $T$ on the training data $X$, denoted as $A = \{\mathbf{x}_1, T(\mathbf{x}_1), \ldots, \mathbf{x}_m, T(\mathbf{x}_m)\} = \{\mathbf{a}_1, \ldots, \mathbf{a}_{2m}\}$, the loss is given by:

$$\ell(a_i, a_j; t) = -\log \left( \frac{\exp\left(\ell(a_i)^T \ell(a_j)/t\right)}{\sum_{k=1}^{2m} 1_{k \neq i} \exp\left(\ell(a_i)^T \ell(a_k)/t\right)} \right) \tag{14}$$

where, $t$ is the temperature that controls the attraction and repulsion forces.

In (Kang & Park, 2021) the discriminator network before the fully connected layer $(D_{\phi 1})$ is considered as the encoder network, an additional multi-layer perceptrons $h$ is used as a projection layer. Instead of using data augmentation, the authors used the embeddings of the class labels to capture the data-to-class relations, the modified loss is given as follows:

$$\ell(x_i, y_i; t) = -\log \left( \frac{\exp(l(x_i)^T e(y_i)/t)}{\exp(l(x_i)^T e(y_i)/t) + \sum_{k=1}^{m} \mathbf{1}_{k \neq i} \cdot \exp(l(x_i)^T l(x_k)/t)} \right) \tag{15}$$

In order to ensure that the negative samples having the same label as $y_i$ are not apart, a cosine similarity of such samples is added to the numerator of Eq. 15 giving rise to the 2C loss:

$$l_{2C}(x_i, y_i; t) = -\log \left( \frac{\exp(l(x_i)^T e(y_i)/t) + \sum_{k=1}^{m} \mathbf{1}_{y_k = y_i} \cdot \exp(l(x_i)^T l(x_k)/t)}{\exp(l(x_i)^T e(y_i)/t) + \sum_{k=1}^{m} \mathbf{1}_{k \neq i} \cdot \exp(l(x_i)^T l(x_k)/t)} \right) \tag{16}$$

where $l(x_i)$ is the embedding of the image $x_i$ and $e(y_i)$ the embedding of the class label $y_i$.

Eq. 16 ensures that a reference sample $x_i$ is drawn closer to its corresponding class embedding $e(y_i)$ while distancing it from other classes.

By minimizing this 2C loss, **ContraGAN** effectively reduces the distance between embeddings of images with the same labels while increasing the distance between embeddings of images with different labels. This dual consideration of data-to-data $l(x_i)^T l(x_k)$ and data-to-class $l(x_i)^T e(y_i)$ relations marks a significant advancement over traditional methods.

### 3.3.2   Rebooting Auxiliary Classifier GAN (ReACGAN)

The introduction of contrastive learning in conditioning GANs paved the way for addressing data-to-data relations, a crucial aspect previously overlooked in previous work, particularly in classifier-based GANs like **ACGAN**. Building on this foundation, Rebooting ACGAN (ReACGAN) (Kang et al., 2021) introduces the **Data-to-Data Cross-Entropy loss (D2D-CE)**. This novel approach specifically targets the early training collapse and the generation quality issues inherent in ACGAN. (Kang et al., 2021) started by considering the empirical cross-entropy loss used in ACGAN, which is given as follows:

$$\mathcal{L}_{CE} = -\frac{1}{N} \sum_{i=1}^{N} \log \left( \frac{\exp(F(x_i)^T w_{y_i})}{\sum_{j=1}^{c} \exp(F(x_i)^T w_j)} \right) \tag{17}$$

where $F : \mathcal{X} \to \mathbb{R}^d$ is feature extractor and a single fully connected layer classifier $C : F \to \mathbb{R}^c$ which is parameterized by $\mathbf{W} = [w_1 \cdots w_c] \in \mathbb{R}^{d \times c}$, where $c$ is the number of classes. (Kang et al., 2021) found that at the early training stage the average norm of ACGAN's input features maps increases. Respectively, the average norm of the gradients dramatically increases at the early training steps and decreases with the high class probabilities of the classifier. In addition, it was observed that as the average norm of the gradients decreases, the FID value of ACGAN does not decrease indicating the collapse of ACGAN.

(Kang et al., 2021) found that normalizing the feature embeddings onto a unit hypersphere effectively solves the ACGAN's early-training collapse. Specifically, the authors of ReACGAN introduced a projection layer $P$ on the feature extractor $F$ and they normalized both the feature embeddings $\frac{P(F(x_i))}{\|P(F(x_i))\|}$ (denoted as $\mathbf{f}_i$ and the weight vector $\frac{w_{y_i}}{\|w_{y_i}\|}$ (denoted as $v_{y_i}$)

In addition to the normalization, (Kang et al., 2021) introduced a contrastive loss **Data-to-Data Cross-Entropy(D2D-CE)** to better capture the data-to-data relations as in ContraGAN, furthermore they introduced two margin values to the D2D-CE to guarantee inter-class separability and intra-class variations. The contrastive **D2D-CE** loss reads:

$$L_{D2D-CE} = -\frac{1}{N} \sum_{i=1}^{N} \log \left( \frac{\exp([f_i^T v_{y_i} - m_p]_- / \tau)}{\exp([f_i^T v_{y_i} - m_p]_- / \tau) + \sum_{j \in N(i)} \exp([f_i^T f_j - m_n]_+ / \tau)} \right) \tag{18}$$

where, $\tau$ is the temperature parameter, and $N(i)$ denotes the set of indices for negative samples with labels different from the reference label $v_{y_i}$ in a batch. Margins $m_p$ and $m_n$ are used to manage similarity values for easy positives and negatives, respectively. The terms $[.]_-$ and $[.]_+$ correspond to the $\min(., 0)$ and $\max(., 0)$ functions.

This contrastive loss function proved to be effective in overcoming the limitations of ACGAN, significantly enhancing both class consistency and image diversity.

In contrast to the **2C** loss, the **D2D-CE** objective does not hold false positives in the denominator, which can cause unexpected repulsion forces. Furthermore, introducing the margins in **D2D-CE** loss can prevent having large gradient that can be caused by pulling easy positive samples.

### 3.4   Towards a unified framework for conditioning the discriminator

As previously discussed, various methods have been introduced to condition the discriminator, either by incorporating auxiliary classifiers or by employing alternative approaches. While the inclusion of a classifier in **ACGAN** effectively achieved conditioning, alternative approaches have successfully conditioned the

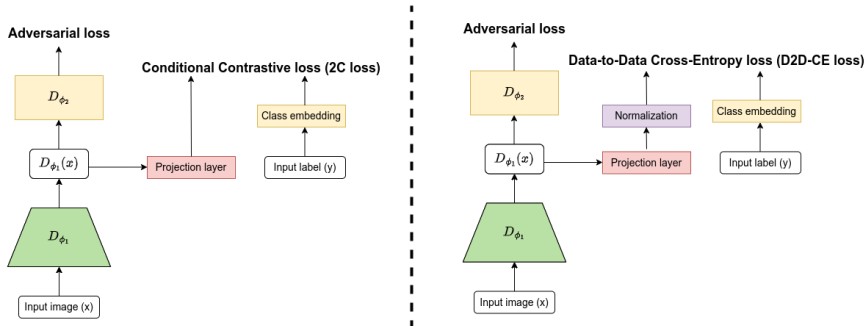

Figure 3: Contrastive learning based architectures. On the left: the discriminator architecture of Contra-GAN. On the right: the Reboot AC-GAN architecture.

discriminator without the need for a classifier. In (Chen et al., 2021), showed that the use of classifiers can benefit conditional generation. Furthermore, they introduced a unifying framework named **Energy-based Conditional Generative Adversarial Networks (ECGAN)** which explains several cGAN variants. In order to connect the classifier-based and the classifier-free approaches used equivalent formulations of the joint probability $p(x, y)$.

From a probabilistic perspective, a classifier can be seen as a function that approximates $p(y|x)$, the probability that $x$ belongs to $y$. Similarly, a conditional discriminator can be viewed as a function that approximates $p(x|y)$, the probability that $x$ is real given a class $y$, the joint probability is given as follows:

$$\log p(x, y) = \log p(x|y) + \log p(y) = \log(y|x) + \log p(x) \tag{19}$$

In Eq.19 we observe that the joint probability distribution $\log p(x, y)$ can be approached through two distinct methods. The first method involves modeling a conditional discriminator $p(x|y)$, while the second focuses on a classifier $p(y|x)$. By sharing the parameterization between these models, the training process becomes mutually beneficial, allowing improvements in the conditional discriminator to enhance the classifier's performance, and vice versa.

### 3.4.1 Approaching the joint distribution from the conditional discriminator perspective

Similar to the energy based models (LeCun et al., 2006), *log p(x, y)* was parameterized in (Chen et al., 2021) using a function $f_\theta(x)$, where $\exp(f_\theta(x)[y]) \propto p(x, y)$.

Therefore, the log-likelihood can be modeled as follows:

$$\log p_\theta(x|y) = \log \left( \frac{\exp\left(f_\theta(x)[y]\right)}{Z_y(\theta)} \right) = f_\theta(x)[y] - \log Z_y(\theta), \tag{20}$$

Where $Z_y(\theta) = \int_{x'} \exp\left(f_\theta(x')[y]\right) dx'$

Optimizing Eq. 20 is intractable because of the partition function $Z_y(\theta)$. By introducing the Fenchel duality of the partition function $Z_y(\theta)$ and a trainable generator $q_\phi(z, y)$, where $z \sim \mathcal{N}(0, 1)$, (Chen et al., 2021) showed that the maximum likelihood estimation in Eq.20 is:

$$\max_\theta \min_\phi \sum_y \mathbb{E}_{p_d(x|y)}\left[f_\theta(x)[y]\right] - \mathbb{E}_{p(z)}\left[f_\theta(q_\phi(z, y))[y]\right] - H(q_\phi(\cdot, y)) \tag{21}$$

By discarding the entropy term, this equation has the form of the GAN.

The discriminator loss in this case is given by:

$$\mathcal{L}_{d_1}(x, z, y; \theta) = -f_\theta(x)[y] + f_\theta(q_\phi(z))[y] \tag{22}$$

Table 2: Some reprsentative cGAN discriminator's architectures as an ECGAN approximation

| Existing cGAN | $\alpha$ | $\lambda_{clf}$ | $\lambda_c$ |
|---|---|---|---|
| ProjGAN | 0 | 0 | 0 |
| AC-GAN | 0 | $> 0$ | 0 |
| ContraGAN | 0 | 0 | $> 0$ |

### 3.4.2 Approaching the joint distribution from the classifier perspective

As depicted in Eq.19, $log\ p(x, y)$ can also be approximated using $log\ p(y|x)$ and $log\ p(x)$. Using the energy function introduced earlier $f_\theta(x)$, $log(p_\theta(y|x)$ can be expressed as:

$$p_\theta(y|x) = \frac{p_\theta(x, y)}{p_\theta(x)} = \frac{\exp(f_\theta(x)[y])}{\sum_{y'} \exp(f_\theta(x)[y'])},\tag{23}$$

Which can be written using a softmax as:

$$\mathcal{L}_{\text{clf}}(x, y; \theta) = -\log\left(\text{SOFTMAX}\left(f_\theta(x)\right)[y]\right)\tag{24}$$

Furthermore, (Chen et al., 2021) showed that by setting $h_\theta(x) = \log \sum_y \exp(f_\theta(x)$, maximizing the log-likelihood of $p(x)$ is equivalent to solving the following optimization problem.

$$\max_\theta \min_\phi \mathbb{E}_{p_d(x,y)}\left[h_\theta(x)\right] - \mathbb{E}_{p(z)}\left[h_\theta(q_\phi(z, y))\right] - H(q_\phi)\tag{25}$$

Similar to Eq. 21, we can see that the Eq. 25 has the form of the traditional GAN. In this case, the equation of the discriminator is given as:

$$\mathcal{L}_{d_2}(x, z, y; \theta) = -h_\theta(x) + h_\theta(q_\phi(z))\tag{26}$$

In (Chen et al., 2021), two approaches were proposed to estimate the entropy terms in Eq. 21 and Eq. 25. The first approach involves considering the entropy term to be zero, based on the fact that entropy is always non-negative, the constant zero is a lower bound. The second approach involves estimating a variational lower bound. The authors demonstrated that the **2C** loss, introduced in ContraGAN, serves as a lower bound in this context.

### 3.4.3 Unifying cGANs discriminators

To unify the classifier-based and classifier-free discriminators, (Chen et al., 2021) proposed the ECGAN discriminator with the following objective:

$$L_D(x, z, y; \theta) = L_{d1}(x, z, y; \theta) + \alpha L_{d2}(x, z, y; \theta) + \lambda_c L_C^{real} + \lambda_{\text{clf}} L_{\text{clf}}(x, y; \theta)\tag{27}$$

where:

$L_{d1}(x, z, y; \theta)$ is designed for the conditional discriminator, adjusting the output specifically for class-corresponding data pairs $(x, y)$. Conversely, $L_{d2}(x, z, y; \theta)$ addresses the unconditional aspect, updating outputs based on the realness of $x$, independent of the class. The classifier training component, $L_{\text{clf}}(x, y; \theta)$, increases the output for the correct class $y$ and decreases it for other classes, thus enhancing classification accuracy. Lastly, the component $L_C^{real}$ which is the contrastive loss calculated on real samples, plays a crucial role in refining latent embeddings, by pulling closer the embeddings of data with the same class. The detailed derivation of this loss can be found in (Chen et al., 2021).

| Method | Method Discriminator Loss | Schematic |
|---|---|---|
| **AC-GAN** | $\min_C \max_D \mathcal{L}_{AC}(GD, C) = \mathbb{E}_{x \sim p_x(x)}[\log D(x)] + \mathbb{E}_{z \sim p_z(z), y \sim p_y(y)}[\log(1 - D(G(z, y)))]$ $- \lambda_c \mathbb{E}_{x \sim p(x,y)}[\log C(x, y)] - \lambda_c \mathbb{E}_{z \sim p_z(z), y \sim p_y(y)}[\log(C(G(z, y), y))]$ |  |
| **TAC-GAN** | $\min_C \max_{D, C^{\mathrm{mi}}} \mathcal{L}_{TAC}(G, D, C, C^{\mathrm{mi}}) = \mathcal{L}_{AC}(G, D, C) + \lambda_{ac} V(G, C^{\mathrm{mi}})$ |  |
| **ADC-GAN** | $\max_{D, C_d} V_{AC}(G, D) + \lambda(\mathbb{E}_{x, y \sim p_{x,y}}[\log C_d(y^+|x)] + \mathbb{E}_{x, y \sim q_{x,y}}[\log C_d(y^-|x)])$ |  |
| **UAC-GAN** | $\min_C \max_{D, T}(D, C, T) = L_{AC}(G, D, C) + \lambda_m V_{MINE}(G, T)$ |  |
| **ProJGAN** | $\max_D \mathbb{E}_{x \sim p(x)}[\log D(x|y)] + \mathbb{E}_{z \sim p(z)}[\log(1 - D(G(z|y)))]$ |  |
| **ContraGAN** | $\min_D l_{2C}(x_i, y_i; t) = -\log\left(\dfrac{\exp(l(x_i)^T e(y_i)/t) + \sum_{k=1}^{m} \mathbf{1}_{y_k = y_i} \cdot \exp(l(x_i)^T l(x_k)/t)}{\exp(l(x_i)^T e(y_i)/t) + \sum_{k=1}^{m} \mathbf{1}_{k \neq i} \cdot \exp(l(x_i)^T l(x_k)/t)}\right)$ |  |
| **ReACGAN** | $\min_D L_{D2D-CE} = -\dfrac{1}{N} \sum_{i=1}^{N} \log\left(\dfrac{\exp([f_i^T v_{y_i} - m_p]_-/\tau)}{\exp([f_i^T v_{y_i} - m_p]_-/\tau) + \sum_{j \in N(i)} \exp([f_i^T f_j - m_n]_+/\tau)}\right)$ |  |

Table 3: A summary of different discriminator conditioning methods, the loss that the discriminators optimize and their schematics

A significant aspect of the **ECGAN** framework presented in this paper is its ability to unify various representative variants of cGAN, including ACGAN, ProjGAN and ContraGAN. These variants are different

approximations within the unified **ECGAN** framework, as shown in Table 2, demonstrating the versatility of **ECGAN**.

# 4 Conditional GAN: conditioning the generator

To condition the generator, most approaches typically involve directly integrating the label with the generator, either through concatenation or by employing some normalization techniques such as **conditional batch normalization** and **adaptive instance normalization** (Odena et al., 2017; Gong et al., 2019; Kang & Park, 2021; Kang et al., 2021; Hou et al., 2021). These methods effectively incorporate label information, enabling the generator to produce outputs that accurately reflect the desired attributes.

In this section, we explore the various normalization techniques used to condition the generator.

## 4.1 Batch Normalization: The Foundation

Batch Normalization (BN) (Ioffe & Szegedy, 2015), originally introduced to improve training stability and convergence speed in deep learning, serves as a cornerstone for many generator conditioning methods. BN standardizes feature maps across the entire batch, normalizing each channel by subtracting the mean and dividing by the standard deviation. It then applies learnable affine transformations to scale and shift the normalized outputs. Formally, given an input batch $x \in \mathbb{R}^{N \times C \times H \times W}$, each batch normalization layer has two learnable parameters, $\gamma_{batch}$ and $\beta_{batch}$ which shift and scale the normalized input, respectively:

$$BN(x) = \gamma_{batch} \left( \frac{x - \mu_c}{\sigma_c} \right) + \beta_{batch} \tag{28}$$

where $\mu_c(x) = \frac{1}{NHW} \sum_{n=1}^{N} \sum_{h=1}^{H} \sum_{w=1}^{W} x_{nchw}$ and $\sigma_c(x) = \sqrt{\frac{1}{NHW} \sum_{n=1}^{N} \sum_{h=1}^{H} \sum_{w=1}^{W} (x_{nchw} - \mu_c(x))^2}$.

Instance Normalization (IN), proposed in (Ulyanov et al., 2017) as an alternative to batch normalization, was motivated by style transfer applications, it is computed as follows:

$$IN(x) = \gamma_{instance} \left( \frac{x - \mu_{nc}}{\sigma_{nc}} \right) + \beta_{instance} \tag{29}$$

where: $\mu_{nc}(x) = \frac{1}{HW} \sum_{h=1}^{H} \sum_{w=1}^{W} x_{nch}$ and $\sigma_{nc}(x) = \sqrt{\frac{1}{HW} \sum_{h=1}^{H} \sum_{w=1}^{W} (x_{nchw} - \mu_{nc}(x))^2}$

In instance normalization, the standard deviation ($\sigma_{nc}(x)$) and mean ($\mu_{nc}(x)$) are computed for each individual instance, whereas in batch normalization (BN), these statistics are computed across the entire batch. In (Ulyanov et al., 2017), it was observed that significant improvements could be achieved using instance normalization.

## 4.2 Conditional Normalization Techniques

In (Dumoulin et al., 2017b) Conditional Instance Normalization (CIN) was introduced to learn different artistic styles with a single conditional network where it takes a content image and a given style to apply and produces a pastiche corresponding to that style. CIN extends IN by making the scaling and shifting parameters condition-dependent:

$$IN(x) = \gamma_{instance}^s \left( \frac{x - \mu_{nc}}{\sigma_{nc}} \right) + \beta_{instance}^s \tag{30}$$

where $s \in \{1, 2, \ldots, S\}$ is randomly chosen from a fixed set of conditions (styles). This was pivotal in learning multiple styles within a single model, as demonstrated in style transfer applications

Similarly, **conditional batch normalization** was used to condition vision systems on text. For instance, it was used in (de Vries et al., 2017) as an efficient technique to modulate convolutional feature maps by text embeddings. Conditional Batch Normalization (CBN) is an extension of the standard batch normalization technique. It introduces the ability to condition the normalization process on external information, such as labels, embeddings, or other auxiliary inputs. By doing so, it allows the network to dynamically adjust its internal feature representation in response to specific conditions.

### 4.3 Adaptive Instance Normalization

In (Huang & Belongie, 2017), **Adaptive Instance Normalization (AdaIN)** was introduced. AdaIN can be seen as an extension of instance normalization, where the shift and the scale are not learnt but computed. Given an input content image $x$ and an input style image $y$ the affine parameters are computed as follows:

$$\text{AdaIN}(x, y) = \sigma_{instance}(y) \left( \frac{x - \mu_{instance}(x)}{\sigma_{instance}(x)} \right) + \mu_{instance}(y) \tag{31}$$

By computing the the affine transformation, AdaIN aligns the channel-wise mean and variance of the input image $x$ to match those of the style image $y$. the authors showed that *AdaIN* lead to better style transfer compared to the other methods. Furthermore, it was extensively used in the StyleGAN family of models (Karras et al., 2020b; 2021)

### 4.4 Feature-wise Linear Modulation

In (Perez et al., 2017), the authors introduced a general purpose method for conditioning a neural network on text embeddings called Feature-wise Linear Modulation (FILM). FILM learns functions f and h which output $\gamma_i, c$ and $\beta_i, c$ to modulate a neural network's activation $F_i, c$, the feature-wise affine transformation is given by:

$$\text{FiLM}(F_{i,c}) = \gamma_{i,c} F_{i,c} + \beta_{i,c} \tag{32}$$

FiLM is computationally efficient as it only requires two parameters per modulated feature map and it does not scale with the image resolution. In BigGAN, a strategy similar to FiLM (Perez et al., 2017) was used to learn the class embeddings.

| Method | Formula | Learnable parameters |
|--------|---------|----------------------|
| **Conditional Batch Normalization** | $CBN(x) = \gamma_{batch}^c \left( \frac{x - \mu_c}{\sigma_c} \right) + \beta_{batch}^c$ | $\gamma_{batch}^c, \beta_{batch}^c$ |
| **Instance Batch Normalization** | $CBN(x) = \gamma_{instance}^c \left( \frac{x - \mu_c}{\sigma_c} \right) + \beta_{inst}^c$ | $\gamma_{inst}^c, \beta_{inst}^c$ |
| **Adaptive Instance Normalization** | $\text{AdaIN}(x, y) = \sigma_{inst}(y) \left( \frac{x - \mu_{inst}(x)}{\sigma_{inst}(x)} \right) + \mu_{inst}(y)$ | None |

Table 4: A summary of the the generator conditioning methods

In Table 4, we summarize the various normalization techniques used for conditioning the generator, highlighting the learnable parameters involved during training.

## 5 Experiments

In this section, we present a comparative analysis of various conditioning techniques across multiple datasets. To ensure fairness and consistency, our evaluation leverages StudioGAN (Kang et al., 2023), a PyTorch-based library that provides comprehensive implementations of numerous GAN architectures and conditioning strategies. The analysis was conducted on CIFAR-10 and a subset of ImageNet (Deng et al., 2009). First, we compared different cGAN models while maintaining their original architectures as described in their respective works. Then, we selected the BigGAN architecture as a reference backbone and applied various discriminator conditioning methods to it.

| Metric | Advantages | Disadvantages |
|---|---|---|
| **FID** | - Correlates well with human judgment.
- Captures both fidelity and diversity.
- Sensitive to mode collapse. | - Requires a large number of samples for accurate results. |
| **Inception Score (IS)** | - Evaluates both diversity and quality of generated samples. | - Does not penalize mode collapse.
- Relies heavily on biases in the pretrained Inception network. |
| **Kernel Inception Distance (KID)** | - More robust than FID for small sample sizes.
- Unbiased and effective for small datasets. | - Computationally expensive for large datasets. |
| **Density and Coverage** | - Separately quantifies fidelity (density) and diversity (coverage). | - Computationally expensive
- Requires a threshold for determining data manifold overlap, which can be arbitrary. |
| **Precision and Recall** | - Balances quality (precision) and diversity (recall). | - Less intuitive compared to FID. |
| **Perceptual Path Length (PPL)** | - Measures latent space smoothness and consistency.
- Highlights interpolation capabilities. | - Does not measure overall quality or diversity. |

Table 5: Advantages and disadvantages of some evaluation metrics

## 5.1 GANs evaluation metrics

The evaluation of generative models has undergone significant transformation to address the diverse challenges posed by evolving architectures and datasets. Early metrics, such as **Mean Squared Error (MSE) and Log-Likelihood**, were primarily used to assess models like Variational Autoencoders (VAEs) (Kingma & Welling, 2019). While these metrics provided insights into reconstruction accuracy and probabilistic modeling, they were limited in their ability to capture perceptual quality, structural coherence, and the diversity of generated samples. These shortcomings highlighted the need for evaluation criteria that better align with human perception. The introduction of GANs in 2014 catalyzed the development of new metrics tailored to generative tasks. The **Inception Score (IS)** (Barratt & Sharma, 2018) was one of the first metrics designed for GANs, measuring the realism and class diversity of generated images by leveraging a pre-trained Inception network. However, IS was criticized for its inability to detect mode collapse effectively and its reliance on class labels, which restricted its applicability to labeled datasets. This limitation paved the way for the **Frechet Inception Distance (FID)** (Heusel et al., 2018), which measures the distance between the distributions of real and generated features in the latent space of an Inception network. FID quickly became the gold standard due to its stronger correlation with human judgment of visual quality and its ability to penalize both lack of diversity and poor realism.

Between 2018 and 2020, the landscape of generative model evaluation continued to evolve, introducing advanced metrics to address specific challenges. **Precision and Recall** (Sajjadi et al., 2018) for Distributions were proposed to separately evaluate the quality (precision) and diversity (recall) of generated samples, providing a more nuanced understanding of model performance. **Kernel Inception Distance (KID)** (Bińkowski et al., 2021) emerged as a robust alternative to FID, particularly for small datasets, due to its unbiased estimation properties and reduced sensitivity to sample size.

To evaluate the smoothness and continuity of the latent space, metrics like the **Perceptual Path Length (PPL)** were introduced (Karras et al., 2020a). PPL measures the perceptual consistency of interpolations in the latent space, which is crucial for applications requiring smooth transitions, such as style transfer and morphing. Additionally, new metrics such as **Coverage and Density** (Naeem et al., 2020) were developed to measure the degree to which the real data distribution is captured by the generative model and the concentration of generated samples within the real data distribution, respectively. These metrics provide deeper insights into mode coverage and overfitting tendencies.

The evolution of these evaluation methods reflects the growing complexity and expectations of generative models. Each metric addresses specific aspects of performance, in Table 5, we summarized the advantages and the disadvantages of some metrics.

In our comparison, we relied on the following set of metrics. Our choice was motivated by the fact that they are widely adopted and commonly used in numerous works to evaluate GANs.

**Inception Score (IS):** The Inception Score (IS) (Salimans et al., 2016; Barratt & Sharma, 2018) is a metric for evaluating the quality of images generated by GANs, leveraging an Inception model (Szegedy et al., 2015) pre-trained on the ImageNet dataset (Deng et al., 2009). It quantifies the performance of GANs based on two criteria: the diversity of the generated images across different classes and the confidence of each image's classification. The score is computed by using the Inception model to predict the class distribution for each generated image, assessing both the individual image clarity through the sharpness of its predicted class distribution and the overall diversity by comparing these distributions across all images.

$$\text{IS}(X_t) = \exp\left(\frac{1}{M}\sum_{i=1}^{M} \text{D}_{\text{KL}}\left(p(y|x_i)\|\hat{p}(y)\right)\right) \tag{33}$$

Where $X_t = \{x_1, \ldots, x_M\}$ is the image samples we target to evaluate.

**Fréchet Inception Distance (FID):** FID (Heusel et al., 2018) is a metric used to evaluate the quality of images generated by GANs. The FID score calculates the distance between the feature vectors of real and generated images, extracted using an Inception model pre-trained on the ImageNet dataset. Specifically, it computes the Fréchet(also known as the Wasserstein-2 distance) between the multivariate Gaussian distributions of the feature vectors of the real and generated images.

$$\text{FID}(X_s, X_t) = \|\mu_s - \mu_t\|_2^2 + \text{Tr}\left(\Sigma_s + \Sigma_t - 2(\Sigma_s\Sigma_t)^{\frac{1}{2}}\right) \tag{34}$$

where $\mu$ and $\Sigma$ are the mean vector and covariance matrix of the features, and the subscripts $s$ and $t$ denote the source and target, respectively.

**Precision and Recall (PR):** Precision and recall metrics serve to evaluate the quality and variety of images produced by generative models, relying on comparisons between the distributions of real and generated images. Precision measures the degree to which the generated images resemble the real data distribution, indicating the accuracy of the images produced. In contrast, recall assesses how well the range of real images is represented within the generated images' distribution, reflecting the model's ability to capture the diversity of the real dataset. Precision and recall are defined as follows:

$$\text{Precision} := \frac{1}{M}\sum_{j=1}^{M} \mathbf{1}_{Y_j \in \text{Manifold}(X_s)} \tag{35}$$

$$\text{Recall} := \frac{1}{N}\sum_{i=1}^{N} \mathbf{1}_{X_i \in \text{Manifold}(X_t)} \tag{36}$$

Where $N$ and $M$ are the number of real and fake samples, the manifolds are defined as:

$$\text{Manifold}(X_1, \ldots, X_N) := \bigcup_{i=1}^{N} B(X_i, \text{NND}_k(X_i)) \tag{37}$$

where $B(x, r)$ is the sphere in $\mathbb{R}^D$ around $x$ with radius $r$. $\text{NND}_k(X_i)$ denotes the distance from $X_i$ to the $k^{\text{th}}$ nearest neighbour among $\{X_i\}$, excluding itself.

**Density and Coverage (PR):**

In (Naeem et al., 2020), it was shown that the process of constructing manifolds using the nearest neighbor function is sensitive to outlier samples, which frequently leads to an overestimated representation of the distribution. To address this overestimation issue, they introduced the Density and Coverage metrics, which correct the problem by incorporating sample counting. These metrics are mathematically defined as follows:

$$\text{Density} := \frac{1}{kM}\sum_{j=1}^{M}\sum_{i=1}^{N} \mathbf{1}_{Y_j \in B(X_i, \text{NND}_k(X_i))} \tag{38}$$

| Method | FID ↓ | IS ↑ | Coverage ↑ | Density ↑ | Recall ↑ | Precision ↑ |
|---|---|---|---|---|---|---|
| AC-GAN | 33.31 ±1.8 | 6.82 ±0.79 | 0.39 ±0.04 | 0.57 ±0.02 | 0.21±0.06 | 0.63 ±0.01 |
| TAC-GAN | 19.32 ±2.58 | 6.36 ±0.06 | 0.60 ±0.04 | 0.79 ±0.01 | 0.40 ±0.01 | 0.70 ±0.02 |
| ADC-GAN | 5.06 ±0.19 | **9.95** ±0.06 | 0.86 ±0.01 | 0.89±0.01 | 0.66 ±0.01 | 0.71 ±0.01 |
| ProjGAN | 32.14 ±1.87 | 7.08 ±0.42 | 0.39 ±0.021 | 0.57 ±0.032 | 0.26 ±0.04 | 0.63 ±0.02 |
| BigGAN | 5.44 ±0.12 | 9.63 ±0.08 | **0.87** ±0.003 | 0.99 ±0.01 | 0.62±0.002 | 0.74 ±0.001 |
| StyleGAN2 | **4.87**±0.18 | 8.1 ±0.02 | 0.86 ±0.01 | 0.82 ±0.01 | **0.68** ±0.01 | 0.7 ±0.00 |
| ContraGAN | 5.99±0.91 | 9.5 ±0.21 | 0.84 ±0.011 | 0.95 ±0.01 | 0.60 ±0.004 | 0.74 ±0.002 |
| Reboot AC-GAN | 5.65 ±0.18 | 9.61 ±0.09 | 0.85 ±0.004 | **0.97** ±0.01 | 0.59 ±0.002 | **0.75** ±0.01 |

Table 6: Best FID score achieved by cGANs architectures during 80000 training steps on the CIFAR 10 dataset. We used the same color for methods within the same group of conditioning techniques: red for methods employing an auxiliary classifier, brown for methods utilizing a projection discriminator, and orange for methods leveraging contrastive learning discriminators.

| Method | FID ↓ | IS ↑ | Coverage ↑ | Density ↑ | Recall ↑ | Precision ↑ |
|---|---|---|---|---|---|---|
| AC-GAN | 117.52 ±4.58 | 9.29 ±0.80 | 0.04 ±0.01 | 0.15 ±0.02 | 0.03±0.05 | 0.15 ±0.034 |
| TAC-GAN | 150.66 ±9.9 | 5.96 ±0.84 | 0.11 ±0.14 | 0.06 ±0.02 | 0.12 ±0.04 | 0.18 ±0.05 |
| ADC-GAN | 15.11 ±1.39 | 15.46 ±0.26 | **0.72** ±0.01 | 0.76 ±0.11 | 0.51 ±0.02 | 0.69 ±0.01 |
| ProjGAN | 181 ±12.88 | 5.46 ±0.85 | 0.02 ±0.01 | 0.084 ±0.02 | 0.07 ±0.06 | 0.25 ±0.00 |
| BigGAN | 44.3 ±7.54 | 11.6 ±0.80 | 0.36 ±0.07 | 0.46 ±0.06 | 0.46 ±0.05 | 0.58 ±0.05 |
| StyleGAN2 | 16.99 ±1.46 | 14.85 ±0.3 | 0.67 ±0.01 | 0.67 ±0.0173 | **0.52** ±0.01 | 0.68 ±0.00 |
| ContraGAN | 22.95 ±3.27 | 12.89 ±0.44 | 0.49 ±0.04 | 0.75 ±0.03 | 0.32 ±0.03 | 0.73 ±0.02 |
| Reboot AC-GAN | **12.55** ±1.3 | **15.93** ±0.1 | 0.70 ±0.04 | **0.99** ±0.05 | 0.33 ±0.02 | **0.8** ±0.02 |

Table 7: Best FID score achieved by cGANs architectures during 80000 training steps on the Carnivores dataset. We used the same color for methods within the same group of conditioning techniques: red for methods employing an auxiliary classifier, brown for methods utilizing a projection discriminator, and orange for methods leveraging contrastive learning discriminators.

$$\text{Coverage} := \frac{1}{N} \sum_{i=1}^{N} \mathbf{1}_{\exists j \text{ s.t. } Y_j \in B(X_i, \text{NND}_k(X_i))} \tag{39}$$

Where k is for the k-nearest neighbourhoods.

## 5.2 Comparing performances of cGANs models

First, we evaluated the performance of eight representative cGAN architectures on the **CIFAR-10** dataset. Table 6 presents the scores achieved across three separate runs for each architecture, reported at 80,000 training steps.

In table 6, we observe that **ACGAN, and ProjGAN** exhibit nearly identical performance. Additionally, these architectures show a tendency to collapse after approximately the first 10,000 training steps, as illustrated in Fig. 4. For the other methods, their performance is relatively similar on this specific dataset, despite belonging to different conditioning families (ADC-GAN, ContraGAN, and StyleGAN2).

Subsequently, in table 7, we evaluated these architectures on a subset of the ImageNet dataset (Carnivores), which consists of larger images sized $128 \times 128$ pixels. We observe that **ACGAN, TAC-GAN, and ProjGAN** struggle to accurately learn the distribution of the Carnivores dataset, as indicated by their high FID values. Furthermore, these models tend to collapse earlier than they do on CIFAR-10, highlighting the challenges they face in scaling to larger images.

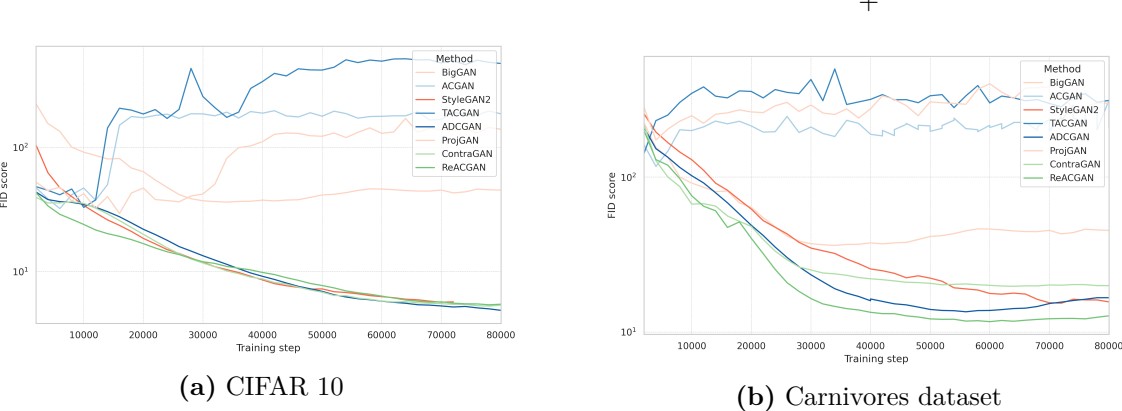

**(a)** CIFAR 10

**(b)** Carnivores dataset

Figure 4: FID scores vs. training steps for different cGAN architectures trained on CIFAR 10 and the Carnivores dataset. We used related colors for methods belonging to the same group of conditioning methods. Namely, blue for methods using an auxiliary classifier, orange for methods using a projection discriminator and green for methods using contrastive learning discriminator. The vertical axis of FID plots is in log scale for better visualization.

In contrast, the other architectures demonstrated strong performance without collapsing. BigGAN achieved a higher FID score, while ReACGAN slightly outperformed the other models. Furthermore, as illustrated in fig 4, these architectures exhibit greater robustness against mode collapse.

| Method | FID ↓ | IS ↑ | Coverage ↑ | Density ↑ | Recall ↑ | Precision ↑ |
|---|---|---|---|---|---|---|
| Big+AC | 39.10 | 18.00 | 0.34 | 0.40 | 0.22 | 0.53 |
| Big+TAC | 102.00 | 10.65 | 0.09 | 0.09 | 0.30 | 0.23 |
| Big+ADC | 41.65 | 17.44 | 0.314 | 0.39 | 0.22 | 0.52 |
| Big+Proj | **32.75** | 18.03 | **0.45** | 0.46 | **0.43** | 0.55 |
| Big+2C | 43.30 | 16.43 | 0.36 | 0.48 | 0.31 | 0.60 |
| Big+DED-CE | 35.77 | **18.22** | 0.38 | **0.49** | 0.19 | **0.59** |

Table 8: Best FID score achieved by various discriminator conditioning methods using a unique backbone on a subset of ImageNet. We used the same colors for methods within the same group of conditioning techniques: red for methods employing an auxiliary classifier, brown for methods utilizing a projection discriminator, and orange for methods leveraging contrastive learning discriminators.

| Method | FID ↓ | IS ↑ | Coverage ↑ | Density ↑ | Recall ↑ | Precision ↑ |
|---|---|---|---|---|---|---|
| Big+AC | 5.51 | 9.86 | 0.85 | 0.94 | 0.66 | 0.74 |
| Big+TAC | 7.01 | 9.80 | 0.82 | 0.83 | **0.66** | 0.71 |
| Big+ADC | 4.93 | **9.88** | 0.82 | 0.90 | 0.65 | 0.72 |
| Big+Proj | 4.89 | 9.82 | 0.86 | 0.94 | 0.64 | 0.73 |
| Big+2C | 5.30 | 9.67 | 0.86 | 0.96 | 0.61 | 0.74 |
| Big+DED-CE | **4.79** | 9.79 | **0.87** | **0.99** | 0.61 | **0.75** |

Table 9: Best FID score achieved by various discriminator conditioning methods using a unique backbone on a subset of CIFAR10. We used the same color for methods within the same group of conditioning techniques: red for methods employing an auxiliary classifier, brown for methods utilizing a projection discriminator, and orange for methods leveraging contrastive learning discriminators.

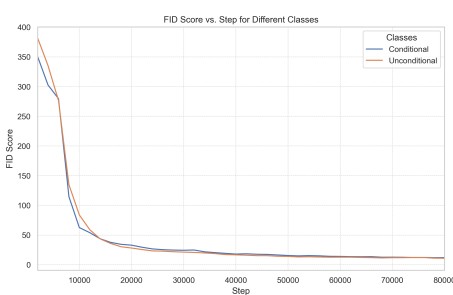

**(a)** Training StyleGAN2 on AFHQ conditionally and unconditionally

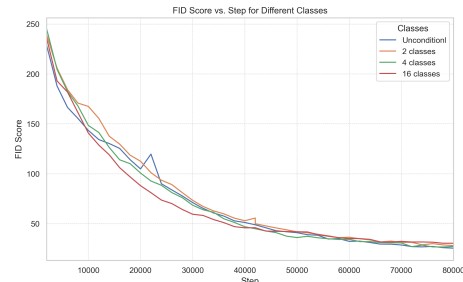

**(b)** Training StyleGAN2 on a subset of ImagNet (vertebrates) with different number of classes

Figure 5: Training StyleGAN2 on AFHQ and a subset of ImageNet (vertebrates) with and without conditioning

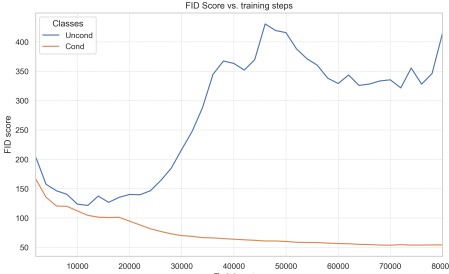

**(a)** Training BigGAN2 on a subset of ImageNet(random) conditionally and unconditionally

**(b)** Training StyleGAN2 on a subset of ImageNet(random) with different number of classes

Figure 6: Training StyleGAN2 and BigGAN on a subset of ImageNet (random) with and without conditioning

### 5.3 Comparing discriminator conditioning methods using a unified backbone

Previously, we compared different cGAN models using their original architectures. However, these cGANs employ various backbones, making it difficult to determine whether their performance is driven by the conditioning method or by other factors related to the backbone. To address this, we adapted the BigGAN backbone to various discriminator conditioning techniques. In Table 9, we observe that when using a unique backbone to condition the GANs on CIFAR10, all the methods converge well. Additionally, AC, Projection, and TAC conditioning do not collapse, indicating that the mode collapse observed in Fig. 4 is likely due to the backbones used in AC-GAN, ProjeGAN, and TAC-GAN, respectively, and not the conditioning techniques themselves. Furthermore, we note that TAC conditioning achieves the highest FID score, while Projection GAN achieves the lowest. In Table 8, we see that using a unified backbone, conditioning the discriminator by projection results in the best FID score among all methods. Additionally, the other methods tend to collapse relatively early, which may be due to the additional components required by these methods, potentially destabilizing the training. In contrast, Projection does not require any additional networks. Surprisingly, when using TAC to condition the network, the GAN fails to achieve a good score. This suggests that TAC may not be compatible with this backbone or that the training becomes more complex due to the twin classifiers.

### 5.4 Conditional vs unconditional image generation

Can we achieve better image generation by providing the network with the class of each image? Intuitively, conditioning the model on class labels can be viewed as providing it with additional information, which has the potential to enhance the quality of the generated images. To investigate this, we trained two StyleGAN2 models using the AFHQ (Karras et al., 2020b) dataset, which comprises three distinct classes. The first model was trained without conditioning (unconditionally), while the second model was trained with conditioning, where class labels were provided. The FID scores for both models are presented in Fig. 5. The results indicate that the two curves are almost similar, suggesting that in this particular case, conditioning does not significantly impact the quality of the generated images.

Given that the AFHQ dataset contains only three distinct classes, we created a specialized dataset derived from ImageNet (we took the superclass vertebrates) to gain a deeper understanding of how varying the number of classes affects model performance. In this dataset, the total number of images (resized to 128x128) was kept constant, while the number of classes was varied. Fig. 5 illustrates the FID score curves for each training scenario. We began with unconditional training and incrementally increased the number of classes (2, 4, 8, 16). Throughout these experiments, the same set of images was used in all training sessions, ensuring that the model consistently learned from the same distribution.

We observe that when we train conditionally on all classes, the convergence rate is slightly faster. This suggests that conditioning can accelerate the convergence process in GANs. To validate this observation, we trained two models, StyleGAN2 and BigGAN, using a subset of 20 random classes from the ImageNet dataset. In Fig. 6, we observe that in both cases, conditioning improves the convergence of the two models. We repeated this experiment with three different subsets of 20 random classes, and the results consistently show that conditional training is faster than unconditional training (the remaining figures are provided in the Appendix B). The effect of conditioning may become more pronounced when the number of classes is larger and the classes are more diverse.

## 6 Conclusion

In this survey, we have explored various methods for conditioning Generative Adversarial Networks (GANs), focusing on three primary families of techniques: Auxiliary-classifier based, Projection based, and Contrastive learning based. Each of these approaches offers unique mechanisms to enhance the control and quality of generated images, addressing different challenges inherent in GAN training.

Auxiliary-classifier based methods, such as AC-GAN and its variants, integrate additional classifiers to improve class-specific image generation. These methods have demonstrated improvements in image quality but often struggle with issues like mode collapse and reduced diversity as the number of classes increases. Subsequent enhancements, like the Twin Auxiliary Classifier GAN (TAC-GAN), have sought to mitigate these issues by refining loss functions and incorporating mutual information estimators.

Projection-based discriminators offer a novel approach by conditioning on the inner product between embedded conditional vectors and feature vectors. This family of methods enhances training stability and performance without requiring additional classifiers. Techniques in this category have proven effective in maintaining inter-class diversity and generating high-fidelity images.

Contrastive learning based methods, exemplified by models like ContraGAN and ReACGAN, address the limitations of earlier approaches by focusing on data-to-data relations. These techniques use contrastive losses to maintain diversity and mitigate mode collapse.

Through a comparative analysis of datasets, including CIFAR-10 and a subset of the ImageNet dataset (Carnivores), we found that enhancements to GANs significantly benefit conditioning methods. Notably, contrastive learning-based architectures, projection-based techniques, and auxiliary classifier methods consistently achieve low FID scores.

In conclusion, the advancements in GAN conditioning techniques have significantly enriched the capabilities of generative models. The insights gained from this body of work are invaluable for guiding future research

and applications in generative modeling. By continuing to innovate and refine these methods, we can unlock new potentials in GANs, paving the way for groundbreaking applications and more controlled, high-quality image generation.

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
