# OpenReview forum: "GANs Class-Conditioning Methods: A Survey"
_TMLR — Rejected by TMLR_

### Review · Reviewer_Ufti · 2024-10-19

**Summary Of Contributions:**

This paper reviews several conditioning methods proposed for GANs. The authors categorize existing conditioning approaches into two main categories: a) discriminator conditioning and b) generator conditioning, and elaborate on notable works within each. The majority of the paper's methodology section is dedicated to explaining various discriminator conditioning approaches (auxiliary classifier-based, projection-based, contrastive learning-based, and unified approaches encompassing these). The authors conduct experiments on selected models using CIFAR-10, a subset of ImageNet (Carnivores), and AFHQ datasets to validate the effectiveness of existing methods. To the best of my knowledge, this is the first survey paper that solely focuses on "conditioning methods" of GANs, explaining notable "class-conditioning" approaches.

**Audience:**

Yes

**Claims And Evidence:**

No

**Requested Changes:**

* The exposition should be improved, including fixing typographical errors and adjusting figure/table placements.
* The authors should revise some claims regarding prior surveys and clarify the scope of their work. There are several existing works that thoroughly analyze GANs (including aspects of conditional GANs), and since this paper focuses solely on class-conditioning in image domains, the authors should explicitly state this.
* As mentioned in the weaknesses, the experimental design and analysis should be more comprehensive. Comparisons between conditioning methods could be improved with better experimental designs. Additionally, the interpretation of the results may be biased due to the use of small datasets and specific models. I expect deeper insights to be supported by more thorough experiments.
* The distinguishing feature of this paper is its focus on the "class-conditioning" aspect of GANs. I suggest that the authors conduct additional experiments to provide more insight into the "conditional" aspect, such as ablation studies of conditioning methods while controlling for other factors in both the generator and discriminator, using a more controlled setup.

**Strengths And Weaknesses:**

**Strengths**:
* The writing (note, not the exposition of the materials) is clear and easy to follow. Even for readers without extensive prior knowledge, I believe the contents of the paper can effectively convey the history and insights of class-conditional GANs. Given that this is a survey paper, this clarity is a solid strength.
* The authors' selection of existing works is consistent and beneficial for explaining the discussed aspects of conditional GANs.

**Weaknesses**:
* The exposition could be improved. There are issues with citations and references (some are marked with "?" and figure/table references are mismatched). Moreover, some figures and tables are located too far from the related text descriptions, making it difficult for readers to follow.
* The experimental designs and results are not sufficiently convincing or comprehensive. As mentioned in the paper, most experiments heavily rely on StudioGAN [3] library. However, this work already provides fully trained checkpoints along with extensive benchmarks on several large-scale datasets, and the main results from this paper do not seem to add significant insights to it. It is mentioned that results are based on 80,000 steps of training, but several methods benefit from longer training periods, and I believe some rankings of models may change as a result. Furthermore, if the authors intended to provide a comparison between class conditioning methods, they should present results using uniform backbones. As several models (e.g., AC-GAN, BigGAN, StyleGAN2) have different backbone architectures, we cannot be certain whether the differences stem from the actual conditioning method or simply a better backbone. Additionally, there are no experiments regarding conditioning methods of the generator.
* Interpretations of the experimental results are often incorrect. While it may be a typo, in Section 4.2, the authors mention that ACGAN, BigGAN, and TAC-GAN exhibit nearly identical performance. However, BigGAN shows superior performance compared to these two. Furthermore, in Section 4.3, it is claimed that conditioning can speed up convergence in GANs, but the training curve suggested in Figure 5-b does not support this claim. Not only does it display just {Uncond, 2, 4, 16} with 8 omitted, but only the variant with 16 classes seems to converge slightly faster than the unconditional variant, and in fact, the unconditional setup ends up with the best FID at the end. As experiments are limited to a subset of ImageNet and the StyleGAN2 architecture, different trends could be observed with more diverse setups. This observation applies to other parts of the experiments as well. I feel that some judgments are based on too small samples and could be potentially biased.
* The paper offers marginal insights. While the explanations provided are clear, an important consideration for a survey paper is the insights that contribute to a better understanding of existing works. I believe deeper insights could be drawn by designing and conducting additional experiments, rather than simply reporting the re-trained metrics of previous works as they are.
* The motivation could be further clarified. On the second page, the authors state that "no prior surveys have comprehensively discussed different architectures used in conditional GANs." As GANs are a well-established topic, several well-known papers [1,2,3] offer insightful experiments and analyses related to conditioning methods and architectures, even if not solely focused on conditional aspects. Additionally, this paper only covers the "class" conditional aspect of GANs. Considering recent trends, it might enhance the paper’s relevance to include a discussion of conditioning in other domains, such as text or audio.

[1] Wang et al., "Generative Adversarial Networks in Computer Vision: A Survey and Taxonomy"

[2] Jabbar et al., "A Survey on Generative Adversarial Networks: Variants, Applications, and Training"

[3] Kang et al., "StudioGAN: A Taxonomy and Benchmark of GANs for Image Synthesis"

---

> ### Author Response · Authors · 2025-01-06
> **Response to Reviewer Ufti**
>
> We sincerely thank the reviewer for their valuable feedback and constructive suggestions, which have significantly contributed to improving the quality of our paper. We are also pleased to know that the reviewer finds our work well-written and the selection of works consistent. Below, we provide detailed responses to the comments and questions raised. To ensure clarity, the modified sections are highlighted in blue in the updated version of the manuscript.
>
> > **Reviewer comment:**
> > The exposition could be improved...
>
> **Response:**
>
> - We apologize for the missing references, we have added all of them, we also improved the exposition of our work in the revised version.
>
> > **Reviewer comment:**
> > The experimental designs...
>
> **Response:**
> - Indeed, we used StudioGAN in our experiments to maintain consistency. We chose to stop training at 80,000 iterations after observing stability in the FID score and slow decrease at this iteration for many models. While, StudioGAN trainning is longer, for some models the best FID is acheived much earlier. Furthermore, retraining the original architectures has allowed us to better track the training dynamics of the models, where we found out that some models tend to mode collapse earlier then other models as shown in **Fig. 5**.
>
> > **Reviewer comment:**
> > The experimental designs...backbone
>
> **Response:**
> - We thank the reviewer for this valuable suggestion. In response, we conducted an experiment where we used the backbone of BigGAN and applied various discriminator conditioning methods. This experiment was performed on two datasets: CIFAR-10 and a subset of ImageNet (comprising 20 randomly selected classes). The table below reports the best FID values obtained from three runs for each configuration. In the revised version of the manuscript, we have also included additional evaluation metrics for a more comprehensive analysis. Our results show that projection-based conditioning achieves the best FID on both datasets. Moreover, other conditioning methods tend to collapse after a certain number of training steps, suggesting that projection-based conditioning provides more robust and consistent performance.
>
> | Method    | CIFAR10    |Imagnet|
> |--------------|--------------|--------------|
> | AC       | 5.51 | 39.10|
> | Proj      | **4.89** | **32.75**|
> | TAC    | 4.93 | 102.0 |
> | ADC   | 7.01 |  41,65|
> | 2C   | 5.28 | 43,3|
> | D2D-CE | 4.79 | 35.77 |
>
> > **Reviewer comment:**
> > Additionally, ... generator.
>
> **Response:**
> - We identified two methods for conditioning GANs: conditional batch normalization (CBN) and adaptive instance normalization (AdaIN). The former is widely used across most conditional GAN architectures, while the latter is predominantly employed in StyleGANs and is not easily applicable to other architectures. For this reason, we restricted our focus to conditional batch normalization in our analysis.
>
> > **Reviewer comment:**
> > Interpretations of the experimental results are often incorrect.
>
> **Response:**
> - We apologize for the typos, we corrected this in the revised version.
>
> > **Reviewer comment:**
> > Furthermore, in Section 4.3,... in GANs.
>
> **Response:**
> - In this experiment, we constructed a specialized subset from ImageNet by leveraging its hierarchical organization. However, this organization can introduce challenges, as some groups of classes may be more homogeneous than others. To better understand the effect of conditioning on training, we created three different datasets, each containing 20 randomly selected classes from Imagnet, and trained StyleGAN2 on these datasets both conditionally and unconditionally. We selected StyleGAN2 for this experiment because it is widely regarded as the state-of-the-art (SOTA) GAN architecture in both conditional and unconditional settings. As shown in Fig. 4, the conditional StyleGAN2 demonstrates faster convergence compared to its unconditional counterpart, highlighting the benefits of conditioning during training. We did also the same experiments with BigGAN as backbone and we observed a more stable tarining with conditioning as shown in **Fig 4**.
>
> > **Reviewer comment:**
> > The paper offer...are clear,
>
> **Response:**
> We apologize for this and hope that the additional experiments enhance the clarity and depth of our paper, making it more insightful.
>
> - While we acknowledge that several well-known papers have conducted experiments and analyses on GANs, our goal is to provide a focused survey specifically on class-conditional GANs and their architectures. Furthermore, our survey aims to explore the theoretical aspects of conditional GANs, establish their connections to other domains, and foster innovation in this field by critically analyzing the limitations of current methods.
>
> - Regarding the focus on class conditioning, we recognize the importance of conditioning in other domains, such as text and audio. To address this, we have added a new subsection that discusses recent advancements in these domains.

---

> > ### Comment · Reviewer_Ufti · 2025-01-17
> >
> > Thanks for updating the paper. I appreciate the new content with clearer designs and additional discussions for generator conditioning.
> >
> > While I acknowledge the academic value in systematically analyzing class conditioning methods in GANs, I share the perspective that the paper's impact could be broadened. Although the work provides a well-structured examination of GAN class-conditioning approaches, its key insights don't substantially advance beyond those found in existing GAN surveys. To enhance the paper's relevance and contribution, I would encourage expanding the discussion to explore how principles from class conditional GANs translate to contemporary research directions (maybe areas where GANs still excel - speed, audio, or discuss hybrid usages with AR or diffusion models). Additional experimental validation or analysis in these aspects would further strengthen the paper's contribution.
> >
> > Nonetheless, I acknowledge the authors' effort in consolidating and organizing the literature on class conditional GANs and appreciate their thorough response to reviewer feedback in improving the manuscript.

---

> ### Author Response · Authors · 2025-01-17
> **Response to Reviewer Ufti**
>
> We thank the reviewer for their positive feedback on the revisions we made.
>
> > **Reviewer comment:**
> > ...its key insights don't substantially ...
>
> **Response:**
>
> Our contributions in this paper can be summarized as follows:
> - We provided a detailed and formal overview of various discriminator class-conditioning methods, categorized into:
>      - Auxiliary-Classifier-Based Discriminators
>      - Projection-Based GANs
>      - Contrastive-Based GANs
>
> Highlighting their motivations and the innovation they brought.
>
> - We discussed the methods  used to condition the generator and found that the majority rely on **Conditional Batch Normalization**. While most existing approaches primarily focus on conditioning the discriminator, this reveals new opportunities to improve conditional GANs by placing greater emphasis on the generator.
>
> - We compared the methods across various datasets. Although a similar comparison was conducted in [3], our study provides deeper insights by tracking the training dynamics of the GANs. Specifically, we observed that certain architectures backbones are more susceptible to mode collapse and struggle to accurately learn the underlying data distribution.
>
> - Additionally, thanks to the reviewer’s suggestion, we performed our comparisons on a unified backbone. We found that projection-based methods outperform other conditioning techniques in terms of quality (measured by FID scores) and stability, demonstrating greater resilience to mode collapse.
>
> - Furthermore, we observed that conditioning significantly enhances convergence in StyleGAN2, which we selected due to its status as a state-of-the-art architecture in both conditional and unconditional generation. Additionally, based on the reviewer’s suggestion, we extended our analysis to another architecture, BigGAN, using different random subsets of the ImageNet dataset. This experiment confirmed the observations we made with StyleGAN2.
>
> We will explicitly include a detailed list of these contributions in our revised version. To the best of our knowledge, these contributions have not been addressed in [1, 2, 3]. Indeed, [1] (2019) is considered one of the earliest surveys on GANs, focusing on the overall architectures used in GANs without a specific emphasis on conditional GANs. The only conditional GAN architectures discussed are the earliest implementations, where the class label is concatenated with the inputs to the discriminator and generator, along with the ACGAN, [2] (2020) also focuses on GANs, the only class-conditional GAN discussed is the AC-GAN. Moreover, no comparative analysis is provided to evaluate the various architectures. Although [3] includes more recent studies, it lacks theoretical explanations of the differences between the class-conditioned architectures and does not compare the conditining techniques on the same backbone. In our revised version, we will clearly highlight the distinctions between our work and these prior studies, emphasizing our unique contributions and addressing these gaps.
>
> > **Reviewer comment:**
> > ...I would encourage expanding the discussion to explore
>
> **Response:**
>
> We thank the reviewer for the suggestion, we have already included a section in the supplementary material where we compare GANs to diffusion models and cited more recent works in GANs that demonstrate their competitiveness in image generation with diffusion models [4]. We can integrate this section into the main text and expand it further in the revised version.
>
> > **Reviewer comment:**
> > ...Additional experimental validation
>
> **Response:**
>
> While additional experimental validation could broaden the impact of our paper, we believe it falls outside the scope of this survey. Our primary goal is to provide a general, concise, yet formal and insightful overview of class-conditional GANs. In this context, we believe our work can serve as a valuable foundation for future research, particularly in developping new class-conditional GANs or combining GANs with other generative models
>
> Regarding inference time, it has been reported in [5] that diffusion models are slower then GANs. Additionally, other models, such as diffusion models and normalizing flows, have demonstrated effectiveness in audio generation, as explored in this comprehensive survey [6].
>
>
> We hope that these updates address the questions raised by the reviewer. If there are any additional concerns, we would be happy to address them.
>
> [1] Wang et al., "Generative Adversarial Networks in Computer Vision: A Survey and Taxonomy"
>
> [2] Jabbar et al., "A Survey on Generative Adversarial Networks: Variants, Applications, and Training"
>
> [3] Kang et al., "StudioGAN: A Taxonomy and Benchmark of GANs for Image Synthesis"
>
> [4] Huang et al., GAN is dead; long live the GAN! A Modern Baseline GAN
>
> [5] Song et al., Denoising Diffusion Implicit Models
>
> [6] Božić el al., A Survey of Deep Learning Audio Generation Methods

---

### Review · Reviewer_QeNF · 2024-12-10

**Summary Of Contributions:**

This paper reviews and performs experimental comparisons on various GAN conditioning methods.

**Audience:**

No

**Claims And Evidence:**

No

**Requested Changes:**

See strengths and weaknesses

**Strengths And Weaknesses:**

1. Structure and Sectioning Issues:
* The structure of the paper seems unbalanced. For instance, Section 2 ("Discriminator Conditioning Approaches") occupies the majority of the paper, while Section 3 is significantly shorter, taking up only about a page.
* Additionally, the terminology in Section 2, such as "discriminator conditioning," is somewhat misleading. All conditioning approaches discussed in the paper condition the generator as well. Without this, conditional generation during inference would not be possible. A more appropriate section title could be "Conditional Training/Losses via Conditional Discriminators."
2. Comparative Table:
* The paper would benefit from the inclusion of a table that compares the various conditioning methods. Suggested columns could include:
    * "Conditioning via Discriminator vs Generator"
    * "Different Losses"
    * "Additional Learnable Parameters"
* Including schematics of the different methods within the table would also enhance clarity and make the comparison more intuitive.
3. Lack of Explanation:
* In Section 2.1.2, which discusses TAC-GAN, the authors mention the missing entropy term but fail to explain how the additional classifier addresses this issue. A more detailed explanation would improve the motivation for the method.
* The same critique applies to Section 2.1.3, where the paper does not fully elucidate how the proposed solution improves upon prior methods.
4. Insufficient Coverage in Section 3:
* Section 3 is too brief and lacks sufficient detail for readers to fully understand the methods discussed. Specifically, it does not address the training aspects of the methods, which is crucial for understanding their applicability and performance. Expanding this section to include more comprehensive coverage of training procedures would significantly improve the paper's utility.
5. Missing References:
* There is a missing reference around the line "Additionally, StyleGAN ?, introduced a style-based generator that...". Please ensure this reference is included to maintain completeness.
* Another missing reference appears around the line: "as seen in Brock et al. (2019); Zhang et al. (2020); Zhao et al. (2020b); Wu et al. (2020); ?". The citation should be added to support the claims made.
* There is a missing reference in the line: "In ?, showed that the use of classifiers can benefit conditional generation. Furthermore, they introduced a unifying framework na...". The relevant reference should be included to support this assertion.
* There are many more throughout the paper.  Please check.

Overall, as a survery paper, I expect a clearer exposition where the readers can more easily extract relevant knowledge.  Also, overall, the papers reviewed are quite a few years old.  By including more recent GAN papers, this paper can become more relevant to the readers of TMLR.

---

> ### Author Response · Authors · 2025-01-05
> **Response to Reviewer QeNF**
>
> We sincerely thank the reviewer for their valuable feedback and constructive suggestions, which have greatly contributed to improving the quality of our paper. Below, we provide detailed responses to the comments and questions raised. For clarity, the modified sections have been highlighted in blue in the updated version of the manuscript.
>
> > **Reviewer comment:**
> > 1.Structure and Sectioning Issues.
>
> **Response:**
> - The structure of the paper reflects recent advances in the conditional GAN literature, which have focused more on the conditioning of the discriminator. This emphasis can be justified by the pivotal role of the discriminator in approximating the distribution of real samples. Indeed, GANs are generative models composed of two neural networks: the generator, which tries to approximate the real distribution of samples, and the discriminator, which tries to distinguish between the real distribution p(x) and the approximated one q(x). GANs are implicit models that do not have access to the model likelihood but instead compare the real and approximated distributions using different strategies, such as comparing the ratios or differences between the two distributions. This focus on discriminators is reflected in the conditional GAN literature.
>
>
> - We thank the reviewer for pointing out this important clarification. We agree that the term "discriminator conditioning" might not fully capture the dual conditioning of both the generator and discriminator, as conditional generation inherently involves conditioning the generator during training and inference as well. To address this, we have revised the section title to "Conditional GAN: Conditioning the discriminator / generator". Additionally, we have clarified in the text that all discussed conditional GANs involve conditioning both the generator and the discriminator . We hope this modification resolves the concern and provides a more precise description of the content.
>
>
> > **Reviewer comment:**
> > 2.Comparative Table
>
> **Response:**
> - We thank the reviewer for this valuable suggestion. In response, we have included a table in Section 2, "Conditional GANs: Conditioning the Discriminator," that provides a summary of the methods discussed, including the method name, the corresponding loss function, and an accompanying schematic for clarity.
>
> - Furthermore, at the end of Section 3, "Conditional GANs: Conditioning the Generator," we have added a table highlighting the conditioning of methods for the generator (which involve different normalization technique ). This table summarizes the normalization methods proposed for conditioning, specifying the method name, the formula for each normalization, and the associated learnable parameters.
>
> > **Reviewer comment:**
> >3. Lack of Explanation:
>
> **Response:**
> - We apologize for the lack of explanation in Sections 2.1.2 and 2.1.3. Regarding TAC-GAN, the authors demonstrated that the absence of the entropy term can lead to a degenerate solution. Estimating this term directly is challenging; to address this, the authors established a connection between this term and the Jensen-Shannon Divergence (JSD) between the conditional distributions $\{Q_{X|Y=1}, \dots, Q_{X|Y=K}\}$, where Q is the target distribution, X the image and Y the label.
>
> - Additionally, they showed that the proposed loss minimizes  the JSD between $\{Q_{X|Y=1}, \dots, Q_{X|Y=K}\}$, as stated in Theorem2 [1], Hence the newly introduced loss function effectively addresses the absence of the entropy term.
>
> - For **UAC-GAN**, the motivation remains the same: minimizing the conditional entropy term. However, instead of directly minimizing it, the authors estimated the mutual information using **MINE** [2]. We have incorporated these clarifications into the revised version.
>
> > **Reviewer comment:**
> >4. Insufficient Coverage in Section 3:
>
> **Response:**
>
> - We appreciate the reviewer’s feedback highlighting the need for more detail in Section 3. In response, we have significantly expanded this section to provide a more comprehensive discussion of these methods and the learnable parameters involved in training.
>
> > **Reviewer comment:**
> >5. Missing References:
>
> **Response:**
> - We apologize for the missing references, we have added all of them.
>
> Despite the emergence of other generative models, such as diffusion models, GANs remain an active area of research [2]. We have added more recent works to the introduction. However, to the best of our knowledge, the most recent works on class-conditioning are those reported in the paper. We hope that our work will inspire further research in this area.
>
> We hope that these updates address all the concerns and questions raised by the reviewer. If there are any additional concerns or questions, we would be happy to provide further clarification.
>
> [1] Gong et al., "Twin Auxiliary Classifiers GAN"
>
>
>
> [2] Shi et al., "On the Analysis of GAN-based Image-to-Image Translation with Gaussian Noise Injection" (ICLR 2024)

---

### Review · Reviewer_5R6t · 2024-12-18

**Summary Of Contributions:**

Survey of the dense thicket of conditional GAN papers on image generation from 2017 to ~2021 (iiuc).

**Audience:**

Yes

**Claims And Evidence:**

No

**Requested Changes:**

Better organization of the material into areas of work, inclusion of audio results, and add a survey of evaluation methods in conditional GANs.

**Strengths And Weaknesses:**

As a survey, I find this a bit lacking, for 3.5 reasons.

First, the review of models reads as a bit of a laundry list. This is a space which has seen an incredible amount of churn, and a survey should help separate papers with signal from the noise. Each model in this list presents some new loss term and some justification, which (per the paper's own evaluation) tend to fail to materialize real results. It would be helpful to group together similar approaches, focusing on those which added truly novel ideas and moved the needle on output quality.

Second, a massive component of the story is entirely missing: Audio generation. People struggled for years to get GANs to produce high-quality audio output. Essentially, you have a spectrogram (or derived features) used as conditioning for audio generation, and want to produce a raw waveform which sounds like the original audio, essentially through synthesizing the phase component of the signal. This is a very tight conditioning problem, which GAN approaches completely floundered on for years. HiFi-GAN provided the major breakthrough (https://arxiv.org/abs/2010.05646), which was eventually realized in the Soundstream Codec (https://arxiv.org/abs/2107.03312). Later work produced EnCodec, which includes some interesting components for balancing the many loss terms involved (https://arxiv.org/abs/2210.13438).

Third, there's a big question of how to know when any of these papers has made real progress. Evaluating generative methods remains quite difficult, and there's basically no discussion about how the evaluation question developed over time, or best practices for separating a good idea from just-another-loss-term.  The paper runs its own evaluations, but this actually limits the paper, IMO: the only systems considered will be ones that play nicely in the image generation framework used by the authors. A better understanding of how evaluation methods have improved (or not!) would help frame the discussion of past methods.

Finally, there's a real question of whether this work is entirely superseded by diffusion methods. It would be interesting to understand where conditional GANs are still considered SOTA relative to diffusion methods.

---

> ### Author Response · Authors · 2025-01-06
> **Response to Reviewer 5R6t**
>
> We sincerely thank the reviewer for their valuable feedback and constructive suggestions, which have greatly contributed to improving the quality of our paper. Below, we provide detailed responses to the comments and questions raised. For clarity, the modified sections have been highlighted in blue in the updated version of the manuscript.
>
> > **Reviewer comment:**
> > First, ...
>
> **Response:**
>
> - Indeed, conditional GANs have witnessed significant advances, and a plethora of methods have been proposed to make the generation process controllable by introducing conditioning. In this survey, we reviewed the major methods that introduced new ideas and established connections between conditional GANs and other areas such as contrastive learning and energy-based models. To this end, we grouped the conditioning methods into two main sections, **Section 2:  Conditional GAN: Conditioning the discriminator** where we discussed the approches to condition the discriminator and **Section 3: Conditional GAN: Conditioning the generator** where we discussed the conditioning methods for the generator.
>
> - In Section 2, we grouped the discriminator conditioning methods into 3 different categories:
>      1. **Auxiliary-classifier-based discriminators (2.1):** In this subsection, we presented methods that use an auxiliary classifier to condition the discriminator, discussed their limitations, and highlighted the improvements proposed to address them.
>      2. **Projection-based discriminators (2.2):** In this subsection, we gave an overview about conditioning the discriminator by projection, we provided  also a list of works that use this method.
>      3. **Contrastive-learning-based discriminators (2.3)** In this subsection, we discussed works that employ contrastive learning for discriminator conditioning.
>     4. **Towards a unified framework for conditioning the discriminator (2.4)**: Finally, we presented a unified perspective on discriminator conditioning through the lens of energy-based models.
>
> - In Section 3, we listed the approches proposed to condition the generator which are mainly based on different conditional normalization techniques.
>   1. **Batch Normalization: The Foundation (3.1):** In this subsection we gave a general introduction to batch normalization.
>   2. **Conditional Normalization Techniques (3.2):** In this subsection we introduced conditional batch normalization that is used for conditioning various conditional GANs.
>   3.  **Adaptive Instance Normalization:** In this subsection we introuced an important technique used in the conditioning of the generator, widely adapted in the StyleGANs family of models.
>
> This organization has been made more explicit in the revised version.
>
>
> > **Reviewer comment:**
> > Second..:
>
> **Response:**
> - We thank the reviewer for highlighting the important and distinct contributions of GANs to audio generation. While our work primarily focuses on class-conditioning methods for GANs, we agree that including this component adds significant value and broadens the impact of our survey. Moreover, we believe that our work can also be of interest to researchers working on audio generation. In response, we have incorporated the suggested works into our paper. These additions provide a broader perspective on the versatility of GANs and their application to challenging tasks like audio generation.
>
> > **Reviewer comment:**
> > Third,...:
>
> **Response:**
>
> - We acknowledge that evaluating generative models, particularly GANs, has been an evolving challenge, and it is important to provide an overview of how evaluation methodologies have progressed over time. To address this, we have added a new subsection to our section GANs Evaluation Metrics, where we summarize the development of these metrics over time. Additionally, we included a table discussing the advantages and disadvantages of each commonly used metric. In our evaluations, we used the most well-known and widely adopted metrics to ensure consistency and relevance.
>
> > **Reviewer comment:**
> > Finally, ...
>
> Indeed, diffusion models have seen significant advancements recently, but they still face limitations, such as long inference times due to the many iterative steps required for sampling[1]. This makes GANs preferable in applications where real-time or low-latency generation is critical. Recent research has explored combining diffusion models with GANs [2] to leverage the advantages of both approaches. In this context, we believe that our survey can provide valuable insights into the various aspects of conditional GANs, serving as a foundation for developing more efficient models.
>
> We hope that these updates address all the concerns and questions raised by the reviewer. If there are any additional concerns or questions, we would be happy to provide further clarification.
>
> [1] Xio et al., Tackling the Generative Learning Trilemma with Denoising Diffusion GANs
>
> [2] Wang et al., Diffusion-GAN: Training GANs with Diffusion

---

> > ### Comment · Reviewer_5R6t · 2025-01-07
> >
> > Thanks for the updates!
> >
> > At a high level, the narrow focus on class-conditioning for image generation still feels quite claustrophobic. Ostensibly, the reason for focusing on image generation is because it's an area where general results should - from a machine learning research perspective - transfer to other domains. However, this empirically has not been the case for GANs, undercutting the notion that this is actually a reasonable foundational research area.
> >
> > * In an era where diffusion models have mainly surpassed GANs for conditioned image generation, we would expect to see tighter focus on either evaluation criteria (eg, latency) or modalities (like audio) where GANs are still worthwhile.
> >
> > * There's apparently no lessons learned from areas outside the class-conditioned image generation (CCIG) scope. Real innovations were required to get GANs working for audio generation, and those innovations are apparently unexamined in the 'core' GAN literature.  One is left to wonder whether other important innovations exist in areas like medical imaging or time series analysis which would translate well to other applications... Perhaps including CCIG.
> >
> > (You could address this problem by better indicating the scope of the survey in the title.)
> >
> > More tactically:
> >
> > 1) Thanks for adding some grouping of methods. In the results section, it would be helpful to indicate which class of methods various models used - it's currently an undifferentiated mass of alphabet soup. This could include a column in the results tables, or using related colors in plots for models from the same family of methods. Alternatively, focusing on the 'best' method from each group and placing full results in supplementary material could help readability.
> >
> > 2) The citations seem to be interspersed with the text with no parentheses, which hurts readabililty.
> >
> > 3) Has anyone run diffusion models through the eval framework you are using? It would be interesting to know how they compare.
> >
> > 4) Should there be a latency or complexity component in the evaluation to differentiate from diffusion models?

---

> ### Author Response · Authors · 2025-01-09
> **Response to Reviewer 5R6t**
>
> We thank the reviewer for their constructive suggestions and insightful questions.
>
> 1. To better indicate the scope of the survey, we changed the title to 'GANs Class-Conditioning Methods: A Survey'
>
> 2. In the tables, we used the same color for methods within the same group of conditioning techniques. In addition, we used related colors for methods belonging to the same groupe in the plots.
>
> 3. We placed all citations in brackets to improve the readability of the paper.
>
> 4. Yes, the metrics we used for the evaluation are commonly employed to assess generative models. For example, in [1], it was shown that diffusion models achieve better results in image synthesis than GANs. In another recent paper [2], the authors proposed a GAN that positively competes against diffusion models. Diffusion models and GANs continue to advance rapidly, competing closely in image synthesis performance.
>
> 5. Inference time is one of the main limitations of diffusion models. To give an order of magnitude, in [3], it was reported that sampling 50k images of size 32 × 32 from a DDPM [4] (a type of diffsuion models) takes around 20 hours, while a GAN can achieve the same task in less than a minute on an Nvidia 2080 Ti GPU. Improving diffusion models inference time remains an active area of research. We have added a section in the supplementary material to better highlight the pros and cons of diffusion models and GANs.
>
> We hope that these updates address the questions raised by the reviewer. If there are any additional concerns or questions, we would be happy to address them.
>
>
> [1] Dhariwal et al., Diffusion Models Beat GANs on Image Synthesis
>
> [2] Huang et al., GAN is dead; long live the GAN! A Modern Baseline GAN
>
> [3] Song et al., Denoising Diffusion Implicit Models
>
> [4] Ho et al., Denoising Diffusion Probabilistic Models

---

### Author Response · Authors · 2025-01-06
**General Response to Reviewers**

We sincerely thank all the reviewers for their thoughtful feedback, constructive suggestions, and valuable comments. Your insights have been instrumental in improving the quality, clarity, and relevance of our paper.

We have carefully addressed all the questions and concerns raised and incorporated the suggested improvements into the revised manuscript. We hope that our responses and updates satisfactorily address all the issues brought up during the review process.

If there are any additional concerns or further clarifications needed, we would be happy to provide them. Once again, thank you for your efforts in helping us enhance this work.

---

> ### Comment · Reviewer_5R6t · 2025-01-06
>
> Hello!
> It seems the version of the paper available in OpenReview is still the original first draft - is it possible to upload the revised version for review?
> Thanks!

---

> > ### Author Response · Authors · 2025-01-07
> > **Response to Reviewer 5R6t**
> >
> > Dear Reviewer 5R6t,
> >
> > Our revised version is available.
> >
> > Best regrads,
> >
> > The Authors

---

### Decision · Action_Editor_a7gd · 2025-01-31

**Recommendation:** Reject

**Comment:**

The authors made a valiant effort to incorporate reviewers' feedback in their submission, in particular with respect to presentation, readability, discussion of evaluation metrics, and the incorporation of additional experiments. Reviewer Ufti acknowledges "the work's contribution in organizing several literatures on class conditional GANs", which "could serve as a useful reference for understanding the evolution of conditioning methods in GANs". Reviewer 5R6t notes that the paper "may be of use to people continuing to work on class-conditioned GANs for images".

Ultimately, however, the submission's strengths do not outweigh the reviewers' concerns over interest to the audience. Refiewer Ufti remains concerned that the works discussed appear "slightly outdated and [have] limited relevance to more recent developments in the field", citing for example one-step generation process (GigaGAN) or adversarial losses for distillation and distribution matching (Adversarial Diffusion Distillation / Distribution Matching Distillation). Reviewer Ufti also remains concerned about the survey's appeal given existing well-established GAN survey papers. Finally, Reviewer 5R6t remains concerned that the submission is "a quite narrowly-scoped survey paper" and is "still borderline for breadth of interest".

**Audience:**

Reviewer Ufti notes the writing quality (aside from the presentation issues) and the fact that to their knowledge the submission is the first survey paper that solely focuses on conditioning methods for GANs.

Reviewers express some concerns over the submission's appeal to the TMLR readership:

- The organization and presentation need improvement. Some references are not resolved (Ufti, QeNF), and additional explanations are needed in Sections 2.1.2 and 2.1.3 to make the material accessible to readers (QeNF). Comparatively less time is spent discussing generator-based conditioning, and more details are needed to help readers understand the approaches discussed in that family of approaches (QeNF).
- The submission is thin on new insights presented and does not contribute enough to a better understanding of existing works (Ufti, 5R6t). Reviewer 5R6t finds the presentation does not do enough to go beyond enumeration and separate the signal from the noise in the literature.
- The fact that the scope is restricted to the image modality (and not text or audio, for instance) limits the paper's interest to the broader community (Ufti, 5R6t). Reviewer 5R6t finds in particular that audio generation is a big omission.
- The papers reviewed are quite a few years old. (QeNF)
- The submission does not present a discussion on evaluation, which remains a challenge to this day (R56t: "A better understanding of how evaluation methods have improved (or not!) would help frame the discussion of past methods.")

**Claims And Evidence:**

Reviewers express some concerns over the claims and evidence:

- The experiments and results are not sufficiently convincing or comprehensive. Reviewer Ufti points out that confounding factors (number of training steps, inconsistencies in backbones used) make it hard to draw definitive conclusions from the results. Reviewer Ufti finds that the main results do not add significant insights on top of the benchmarking results presented by the StudioGAN library the paper uses. Reviewer Ufti is also skeptical that the observations generalize beyond the limited settings in which the experiments are presented.
- Claims made are not always supported by the empirical results. Reviewer Ufti points out that (i) despite the paper's claims in Section 4.2, Ac-GAN, BigGAN, and TAC-GAN do not exhibit nearly identical performance; and (ii) the training curve in Figure 5b does not support the claim that conditioning can speed up convergence in GANs.
- Reviewer Ufti pushes back on the claim that "no prior surveys have comprehensively discussed the different architectures used in class-conditional GANs", citing Wang et al.'s "Generative Adversarial Networks in Computer Vision: A Survey and Taxonomy", Jabbar et al.'s "A Survey on Generative Adversarial Networks: Variants, Applications, and Training", and Kang et al.'s "StudioGAN: A Taxonomy and Benchmark of GANs for Image Synthesis" which, while not specifically focused on conditional GANs, do discuss them.
- Reviewer QeNF pushes back on the term "discriminator conditioning approaches", as the generator still needs conditioning in these approaches.

**Resubmission Of Major Revision:**

The authors may consider submitting a major revision at a later time.